# OpenReview forum: "Dual Mechanisms of Value Expression: Intrinsic vs. Prompted Values in LLMs"
_ICLR.cc/2026/Conference — ICLR 2026 Conference Desk Rejected Submission_

### Official Review · Reviewer_NTL2 · 2025-10-27

**Soundness:** 4
**Presentation:** 4
**Contribution:** 3
**Rating:** 6
**Confidence:** 4

**Summary:**

This paper aims to analyze the internal differences between LLMs with and without prompts using an interpretable approach. By combining Concept Activation Vectors (CAVs) and MLP neuron analysis, the authors investigate how prompting affects internal neuron activations and conceptual representations.

**Strengths:**

1. The paper presents an interpretable approach to analyzing the internal differences between no-prompt and prompted model behaviors. The integration of CAVs with MLP neuron analysis is well executed and conceptually appealing—using CAVs to interpret neuron activations is both novel and insightful.
2. The paper offers concrete takeaways that deepen our understanding of how adding prompts alters internal representations in LLMs, which I believe is beneficial to the broader research community.

**Weaknesses:**

The experiments mainly rely on value vectors to verify hypotheses, while value neurons are only used as motivation. I would have liked to see experiments exploring which concepts the shared neurons correspond to, and what kinds of meanings are captured by unique neurons (intrinsic) and unique neurons (prompted).
Recent works have shown progress in automatically generating neuron explanations [1][2]. The absence of such analysis here is somewhat disappointing — if the authors could include experiments visualizing or describing these neuron-level concepts, I would consider increasing my score.

**Questions:**

1. Why is the value vector computed by averaging and subtracting embeddings? As far as I know, one could directly train a linear classifier and use its normal vector as the Concept Activation Vector (CAV). Wouldn’t this yield a more principled representation that can still be used for identifying value neurons?
2. Line 297: “This may partly reflect limitations of our win rate metric, which compares steered outputs to strong original responses (e.g. Benevolence), making large improvements inherently difficult to observe.” Could this limitation be observed empirically? I’m also curious why the metric drops after Layer 20 — if large improvements exist, would they be noticeable through human evaluation?
3. Line 367: “Previous experiments suggest that shared components are crucial for expressing values.” Which previous experiments does this refer to? Any inference or reference?

[1]https://openaipublic.blob.core.windows.net/neuron-explainer/paper/index.html
[2]The Importance of Prompt Tuning for Automated Neuron Explanations NIPSW

---

> ### Author Response · Authors · 2025-11-22
> **Official Comment by Authors (Part 1/3)**
>
> Thank you for your time and effort for reviewing our work. We sincerely appreciate your comments highlighting the novelty and insightfulness of our neuron-level analysis using Concept Activation Vectors (CAVs), as well as your view that our findings offer concrete insights into how prompts shape LLM representations and benefit the broader research community. We are also grateful for your constructive suggestions on additional neuron-level analyses that could further strengthen our research.
>
>
> > **W1. The experiments mainly rely on value vectors to verify hypotheses, while value neurons are only used as motivation. I would have liked to see experiments exploring which concepts the shared neurons correspond to, and what kinds of meanings are captured by unique neurons (intrinsic) and unique neurons (prompted). Recent works have shown progress in automatically generating neuron explanations. The absence of such analysis here is somewhat disappointing — if the authors could include experiments visualizing or describing these neuron-level concepts, I would consider increasing my score.**
>
> **A.** Thank you for your suggestions! This is a great idea, so we conducted additional experiments based on [1] and [2] to analyze the specific roles of the neurons that we identified.
>
> Specifically, we fed a large corpus (50,000 random excerpts of 64 tokens from OpenWebText [3]) through the model and recorded the top-10 highest activating excerpts for each neuron. After that, we provided those examples to an explainer model (GPT-4o-mini) based on the summary prompt format from [2]. For each of the ten Schwartz values, we selected, from each group (shared, unique‑intrinsic, unique‑prompted), the neurons with the largest L2 norm in the SVD subspace defined in our paper (see Section 2.2), and inspected the automatically generated explanations.
>
> Surprisingly, we found that many of those neurons correspond to meaningful and interpretable features. We have submitted an updated version of the manuscript (see Section 4.1, 4.2, and Appendix K) that includes an extensive list of these neuron-level descriptions for all models. A concise summary of the representative patterns for Qwen2.5-7b-Instruct model is provided below.
>
> **Shared neurons:**
> These consistently encode abstract, high-level features central to each value.
> For example,
> - a Tradition shared neuron at [Layer 14, Neuron 587] fires on mentions of spiritual or religious practices and group rituals;
>
> - The Conformity/Power shared neuron [L11, N15699] fires to social approval, sanction, and collective restraint language;
>
> - For Security, a shared neuron at [L14, N817] responds to institutional risk or safety concepts (like “overload”, “shattering”, “emergency”);
>
> - For Universalism, a neuron at [L13, N1954] is activated by societal ideals and collective welfare;
>
> - Benevolence, [L12, N2456], responds to positive social impact and community well-being.
>
> In all cases, these shared neurons map closely onto the core semantics of each value in Schwartz’s theory. Importantly, they do not merely respond to specific keywords; they generalize to diverse linguistic realizations of each value.
>
> **Prompted-unique neurons:** These fire most strongly for explicit value definitions and keywords.
>
> - For Security, the prompted-unique neuron [L14, N4228] activates on explicit terms like “danger”, “warning”, “threat”;
> - In Achievement, [L12, N7214] is responsive to “growth”, “overcoming”, “improvement”, etc.;
> - For Tradition, [L13, N1047] activates for “heritage”, “legacy”, and “preservation”;
> - In Stimulation, [L13, N3872] is triggered by “adventure”, “thrill”, “exciting”, and similar keywords.
>
> Prompted-unique neurons tend to activate for specific value-related keywords or phrases. We speculate that this arises because prompted behaviors are guided by value-inducing system instructions, causing the model to focus on terms and expressions closely associated with the prompted value. This pattern is consistent with our broader observations: when using prompted value vectors for steering, the model has stronger tendencies to repeat specific keywords (Section 3.2.2, Section 4.2).

---

> ### Author Response · Authors · 2025-11-22
> **(Part 2/3)**
>
> **Intrinsic-unique neurons**:
> These respond to contextual signals and scenario features that tend to co-occur with the value in natural language.
> - For Achievement, an intrinsic-unique neuron [L12, N8187] fires for contexts mentioning personal projects, overcoming setbacks, and challenging circumstances, even if the value is not named;
> - Universalism’s intrinsic-unique neuron [L13, N3111] activates for scenarios involving community, collaboration, or diverse groups;
> - In Hedonism, [L13, N1950] responds to contexts featuring group leisure, food, or enjoyment;
> - For Tradition, [L13, N2197] responds to specific community events or festivals..
>
> These neurons appear to function as contextual cue detectors, responding to situations and environments likely to evoke the target value—even when the value itself is not explicitly mentioned. This enables the model to recognize and express values across a wide range of scenarios and linguistic patterns, rather than relying solely on explicit keywords. We believe this property of intrinsic-unique neurons contributes to greater diversity in value expression, as the model can leverage richer contexts and a broader vocabulary associated with each value. This observation aligns with our findings in Section 3.2.2 and Section 4.2, where we show that intrinsic-unique components promote lexical diversity precisely because they are not limited to specific keywords.
>
> It is important to note, however, that not all neurons exhibited such clear and interpretable patterns. While a substantial number of neurons in Qwen2.5-7B-Instruct could be meaningfully characterized as described above, we also observed some neurons that fired strongly for seemingly random or idiosyncratic contexts. Moreover, this interpretability was less pronounced in other models, such as Llama-3.1-8B and Qwen2.5-1.5B, where neuron-level explanations were often more diffuse or ambiguous. We have provided a more detailed analysis, along with illustrative examples for each model, in the updated manuscript (Appendix K).
>
> [1]https://openaipublic.blob.core.windows.net/neuron-explainer/paper/index.html
>
> [2]The Importance of Prompt Tuning for Automated Neuron Explanations NIPSW
>
> [3]https://huggingface.co/datasets/Skylion007/openwebtext
>
>
> > **Q1. Why is the value vector computed by averaging and subtracting embeddings? As far as I know, one could directly train a linear classifier and use its normal vector as the Concept Activation Vector (CAV). Wouldn’t this yield a more principled representation that can still be used for identifying value neurons?**
>
> **A.** Thank you for the insightful question. You are absolutely right that training a linear classifier and using its normal vector as a Concept Activation Vector (CAV) [4] is a principled and widely adopted approach for probing value directions in neural representations. In fact, we also experimented with this method. However, while the classifier-based vectors achieved reasonable classification accuracy, we found that they were less effective for steering the model’s outputs in practice.
>
> Additionally, given the scale of our dataset (150,321 responses over 26,334 queries), the difference-in-means method offered a much simpler and more computationally efficient solution. It allowed us to estimate value vectors quickly and robustly, without the need for iterative classifier training or hyperparameter tuning. For these reasons (both practical and conceptual), we chose the difference-in-means approach as our main method for identifying value directions.
>
> [4] Been Kim, Martin Wattenberg, Justin Gilmer, Carrie Cai, James Wexler, Fernanda Viegas, et al. 2018. Interpretability beyond feature attribution: Quantitative testing with concept activation vectors (tcav). In International conference on machine learning, pages 2668–2677. PMLR.
>
> > **Q2-1. Line 297: “This may partly reflect limitations of our win rate metric, which compares steered outputs to strong original responses (e.g. Benevolence), making large improvements inherently difficult to observe.” Could this limitation be observed empirically?**
>
> **A**. Thank you for the clarifying question about Line 297. To examine this limitation empirically, we analyzed the Pearson correlation between the value-alignment score of the unsteered model and the delta score (steered minus unsteered) in the PVQ experiments. For a more reliable correlation analysis, we also included additional models beyond those in the main paper (Gemma2-9B-IT, Qwen2.5-32B, Qwen3-8B, Qwen3-14B).
>
> We observed strong negative correlations: intrinsic steering (r = –0.6605, p < 0.001) and prompted steering (r = –0.7443, p < 0.001). These results indicate that when the original responses are already well aligned with the target value dimension, the improvement from steering tends to be smaller.

---

> ### Author Response · Authors · 2025-11-22
> **(Part 3/3)**
>
> The following is an example of the baseline and steered response on the benevolence value dimension in the situational dilemmas dataset:
>
> **Situation:** *Emily is a team leader at a software development company. Her team has recently hit a snag with a project that's nearing its deadline. The stress levels are high, and overtime seems like the only solution to meet the deadline. One of her team members, Jake, has been struggling with his tasks due to personal issues at home. Emily is aware that forcing him to work overtime could exacerbate his problems.*
>
> **Question:** *Should Emily ask Jake to work extra hours to ensure the project is completed on time, or should she allow him the space he needs to manage his personal challenges?*
>
> **Unsteered response:**
> *Emily should have a private conversation with Jake to understand the extent of his personal issues and discuss potential solutions. She could offer Jake flexible working hours, allow him to work from home, or redistribute his workload among the team. Additionally, Emily should consider extending the project deadline if possible, to ensure that all team members can work without undue stress.*
>
> **Intrinsic-steered response:**
> *Emily should not ask Jake to work extra hours. Instead, she can:*
> - *Offer flexible working arrangements to help him manage his personal and professional responsibilities.*
> - *Provide support and resources, such as counseling or time off, to help him through this challenging period.*
> - *Reassess the project timeline and workload distribution to ensure all team members can meet the deadline without compromising their health and well-being.*
>
> This example shows that steering induces a clearer expression of the benevolence value, yet the resulting delta remains small because the baseline response already emphasizes benevolence.
>
>
> > **Q2-2. I’m also curious why the metric drops after Layer 20 — if large improvements exist, would they be noticeable through human evaluation?**
>
> **A.**  Thank you for your insightful question. As noted in Figure 4, we consistently observe the steering metric drops after layer 20 in the PVQ (Portrait Value Questionnaire) setting. While the exact reason remains somewhat uncertain, one of our hypotheses is that this is primarily due to the interaction between the PVQ task format and how late-layer value vectors operate.
>
> Specifically, the PVQ is a multi-choice questionnaire where the model must select among fixed options, rather than freely generate value-laden language. Our lexical analysis (see Section 4.2 and Figure 8) shows that, in late layers, value vectors increasingly promote specific tokens closely associated with a value (e.g., `success' for Achievement). As a result, steering late layers appears to bias the model toward outputting those value-related keywords, which may actually hinder performance on a multiple-choice task where selecting the correct Likert option is required. In other words, while middle-layer steering effectively modulates the model's abstract value state, late-layer steering overconstrains the output distribution, sometimes interfering with the task’s specific output requirements.
>
> Regarding your second question, if we understand it correctly, you are asking whether improvements in value expression, even in cases where the metric drops, would still be detectable through human evaluation. If so, we agree: human evaluation can often capture nuanced improvements in value expression or alignment that may not be fully reflected in task-specific metrics. If this is not what you intended to ask, please let us know, and we would be happy to address it in more detail.
>
> > **Q3. Line 367: “Previous experiments suggest that shared components are crucial for expressing values.” Which previous experiments does this refer to? Any inference or reference?**
>
> **A.** "Previous experiments" in this sentence refer to the results of Table 1 and Figure 5. We have updated this section in the revised version of the manuscript as follows:
>
> Originally, the sentence read:
> > *“Previous experiments suggest that shared components are crucial for expressing values. At the vector level, ……”*
>
>  In the revised version, we now write:
> > *“Our steering experiments on PVQ and situational dilemmas (Table 1 and Figure 5) indicate that shared components are crucial for expressing values. At the vector level, ……”*

---

> > ### Comment · Reviewer_NTL2 · 2025-11-27
> >
> > For Q2-1, thanks for the examples. In those cases, the **original responses** were already quite strong, and after steering it just seemed to add some instruction-following behavior, without making a big difference in the value expression itself.
> >
> > For Q2-2, what I meant is exactly what you described: “Even if the metric goes down, aren’t there still real improvements in value expression from a human perspective?” I mainly wanted to see some examples—what do the responses look like when the metric drops after steering beyond layer 20? I feel that for value expression, relying only on a metric is too rigid, and concrete examples give a much more intuitive sense.
> >
> > Thanks for adding the extra experiments. It’s clear that **prompted-unique neurons** correspond to more specific keywords, while **intrinsic-unique neurons** capture more abstract contextual cues. I’ll raise my score to 8.

---

> > > ### Author Response · Authors · 2025-12-03
> > > **Response to your follow-up regarding Q2-2**
> > >
> > > Thank you very much for the clarification and for your thoughtful follow-up.
> > >
> > > >**For Q2-2, what I meant is exactly what you described: “Even if the metric goes down, aren’t there still real improvements in value expression from a human perspective?” I mainly wanted to see some examples—what do the responses look like when the metric drops after steering beyond layer 20? I feel that for value expression, relying only on a metric is too rigid, and concrete examples give a much more intuitive sense.**
> > >
> > > Regarding Q2-2, we completely agree that relying only on the PVQ metric can be too rigid, and that concrete examples are crucial for understanding value expression. To address your question directly, here is an example from our situational dilemmas dataset for Hedonism, using a prompted value vector applied at layer 24 (a layer where the PVQ metric has already started to decrease):
> > >
> > > **Base response (no steering)**
> > > >*Decision: Alex should prioritize the work project but consider alternative ways to fulfill the travel dream. Perhaps Alex can discuss the situation with the boss to see if there is a possibility of delegating some tasks or working on the project remotely for a short period. Alternatively, Alex could save the travel funds and plan a trip to Brazil at a later time when both personal and professional commitments are more manageable.*
> > >
> > >
> > > **Steered response (prompted Hedonism value vector, layer 24)**
> > > >*Decision: Alex should seize the chance for the Amazon rainforest adventure, but with a twist. Instead of rushing through the experience, Alex can plan to savor each moment, perhaps even sharing the joy with a colleague or friend. To ensure the work project remains a priority, Alex can set small, achievable goals that can be accomplished in between the adventures. This way, Alex can indulge in the thrill of the trip while still delivering on the project's demands.*
> > >
> > >
> > > In the PVQ setup, steering with the same Hedonism vector lowers the multiple-choice score relative to mid-layer steering, because late layers increasingly bias the model toward specific value-related keywords rather than the exact Likert label. However, in this open-ended dilemma, a human reader would likely perceive the steered response as much more hedonism-oriented: it not only chooses the more pleasure-focused option (taking the trip) but also uses value-typical language such as rejuvenate the senses', cherished memories', disconnect from the daily grind', and indulge in the thrill of the trip'.

---

> > > ### Author Response · Authors · 2025-12-03
> > > **Thank you for your insights and score update.**
> > >
> > > Thank you for your positive feedback and for raising the score to 8.
> > >
> > > We are delighted that the additional experiments and qualitative examples addressed your questions, and we appreciate your time and support throughout the review process.

---

### Official Review · Reviewer_cWMP · 2025-10-31

**Soundness:** 3
**Presentation:** 3
**Contribution:** 3
**Rating:** 8
**Confidence:** 4

**Summary:**

This paper investigates how LLMs express values intrinsically (from training) versus via prompts. It extracts “value vectors” from residual stream activations and attributes them to “value neurons” in MLP layers, finding overlapping but distinct mechanisms. Prompted mechanisms are more steerable; intrinsic ones yield greater lexical diversity. After orthogonalizing shared components, intrinsic-unique parts still drive diversity, while prompted-unique parts behave like instruction-following mechanisms, even influencing distant tasks (e.g., jailbreak susceptibility). Experiments span Qwen and Llama-3 models and multiple languages using PVQ questionnaires and situational dilemmas, with GPT-4o-mini as an evaluator. Overall, intrinsic and prompted values are mechanistically separable.

**Strengths:**

This paper presents a novel mechanistic analysis of LLM values, in contrast to most related papers, which conduct only behavioral evaluations.

Interesting empirical findings, such as prompted > intrinsic in steerability; intrinsic > prompted in lexical diversity; unique prompted components align with instruction following.

This paper presents a broad, multilingual evaluation using PVQ and situational dilemmas across several open models, as well as robustness analyses across layers.

It holds practical implications for alignment and persona steering (e.g., steerability vs. diversity trade-offs; jailbreak effects).

**Weaknesses:**

I am particularly interested in how we can leverage these findings to align LLM values. It would be beneficial to compare steering LLM value activations with behavioral alignment in terms of their effectiveness in aligning LLMs.

Linear difference-in-means vectors and neuron projections may miss nonlinear mechanisms. The preconceptualization of the intrinsic value mechanism requires further validation.

**Questions:**

How does value steering based on your discovery compare against behavioral alignment (e.g., RLHF and DPO) in aligning LLM values?

---

> ### Author Response · Authors · 2025-12-03
> **Official Response to Reviewer cWMP (Part 1/2)**
>
> We appreciate your positive assessment regarding the novelty and thoroughness of our mechanistic analysis of LLM values. We specifically thank you for acknowledging our empirical findings, notably that prompted mechanisms exceed intrinsic mechanisms in steerability, while intrinsic surpass prompted mechanisms in lexical diversity. We find your suggestions on the application of our findings to value alignment particularly valuable and believe they offer a promising direction for further development.
>
> > **W1: I am particularly interested in how we can leverage these findings to align LLM values. It would be beneficial to compare steering LLM value activations with behavioral alignment in terms of their effectiveness in aligning LLMs.**
>
> **A1.**
> We can answer this question from two perspectives: (1) aligning pluralistic values, and (2) alignment effectiveness.
>
> **Value Steering as a Means to Align Pluralistic Values.**
> In many AI alignment settings, researchers have selected specific values (typically helpfulness, harmlessness, and honesty) and represented this “collective value” as a **single scalar** reward or preference. However, many researchers have pointed out that real-world human values are far more diverse, inherently pluralistic, and sometimes conflicting (e.g., safety vs. freedom). In response, they have recently sought to incorporate more diverse and pluralistic values into AI systems [1, 2].
> One challenge here is how to align LLMs with multiple values simultaneously, since most alignment algorithms rely on a single scalar value score. As a result, value steering has garnered significant attention as an effective inference-time solution for steering a trained model toward specific values. This method is both flexible, as it can control multiple values simultaneously, and more cost-efficient than full training.
>
> **Alignment Effectiveness.**
> Prior study has shown that activation engineering achieves better performance on benchmarks like TruthfulQA compared to standard SFT and RLHF baselines [3]. Interestingly, activation engineering could be applied on top of standard fine-tuning, which a prior study demonstrated can achieve better out-of-distribution generalization than basic SFT [4].
> Although how well value steering can preserve the model's general capabilities is an important research topic worth investigating, its applicability to pluralistic value alignment and its effectiveness position this method as a promising direction for future development. To develop a more robust theoretical foundation, further research is needed to analyze the behavior and underlying mechanisms, which our work aims to advance.
>
> ###### [1] Sorensen, T., Moore, J., Fisher, J., Gordon, M. L., Mireshghallah, N., Rytting, C. M., ... & Choi, Y. (2024, July). Position: A Roadmap to Pluralistic Alignment. In International Conference on Machine Learning (pp. 46280-46302). PMLR.
>
>
> ###### [2] Rozen, N., Bezalel, L., Elidan, G., Globerson, A., & Daniel, E. Do LLMs have Consistent Values?. In The Thirteenth International Conference on Learning Representations.
>
> ###### [3] Li, K., Patel, O., Viégas, F., Pfister, H., & Wattenberg, M. (2023). Inference-time intervention: Eliciting truthful answers from a language model. Advances in Neural Information Processing Systems, 36, 41451-41530.
>
> ###### [4] Rimsky, N., Gabrieli, N., Schulz, J., Tong, M., Hubinger, E., & Turner, A. (2024, August). Steering llama 2 via contrastive activation addition. In Proceedings of the 62nd Annual Meeting of the Association for Computational Linguistics (Volume 1: Long Papers) (pp. 15504-15522).

---

> ### Author Response · Authors · 2025-12-03
> **(Part 2/2)**
>
> > **W2: Linear difference-in-means vectors and neuron projections may miss nonlinear mechanisms. The preconceptualization of the intrinsic value mechanism requires further validation.**
>
> **A2.** We acknowledge that the linearity of value representations is an important assumption in our work. This is grounded in the linear representation hypothesis, which proposes that simple features and concepts are represented as linear directions within the model’s activation space [5].
> Recent studies strongly support this view, showing that high-level features, such as human values, personality traits, political orientation, and emotions can be identified and steered via linear directions in activation space [6][7][8][9]. We build upon this literature and empirically show, through our thorough evaluation protocols (i.e., PVQ, Situational Dilemma, Value Portrait benchmark) that human values can be identified by these linear directions. The fact that our linear steering vectors consistently change model behavior across these benchmarks shows empirical evidence that value representations can be steered via linear directions in activation space. One possible explanation for the plausibility of linear value vectors is that values are associated with somewhat distinct vocabularies. We can therefore think of specific directions in hidden states as promoting the logits of value-associated vocabularies, although this hypothesis is not theoretically guaranteed yet and is worth further investigation.
>
> To further examine the linearity assumption, we followed the methodology of [10] which compares linear ridge probes with substantially more expressive two-layer MLP probes. Motivated by their analysis, we ran a similar comparison on our own representations. On a random 10% subsample of the situational-dilemma extraction set, we trained both linear probes and two-layer MLP probes (with a hidden layer of size 256) on residual-stream activations to predict which Schwartz value is expressed. Linear probes already achieve high predictive performance (F1 ≈ 0.952), and MLP probes yield only a modest improvement (F1 ≈ 0.975). The small gap between linear and nonlinear probes suggests that the relevant value information is largely linearly decodable in the residual stream, even though additional nonlinearity can provide a slight benefit. Taken together, (i) prior work on linear representations, (ii) our steering experiments showing linear scaling of value scores with vector strength, and (iii) our probe comparison in the situational-dilemma setting all point to value-related features being well-approximated by linear directions in activation space.
>
> ###### [5] Park, K., Choe, Y. J., & Veitch, V. (2024, July). The Linear Representation Hypothesis and the Geometry of Large Language Models. In International Conference on Machine Learning (pp. 39643-39666). PMLR.
>
> ###### [6] Kim, J., Evans, J., & Schein, A. Linear Representations of Political Perspective Emerge in Large Language Models. In The Thirteenth International Conference on Learning Representations.
> ###### [7] Chen, R., Arditi, A., Sleight, H., Evans, O., & Lindsey, J. (2025). Persona vectors: Monitoring and controlling character traits in language models. arXiv preprint arXiv:2507.21509.
> ###### [8] Konen, K., Jentzsch, S., Diallo, D., Schütt, P., Bensch, O., El Baff, R., ... & Hecking, T. (2024, March). Style Vectors for Steering Generative Large Language Models. In Findings of the Association for Computational Linguistics: EACL 2024 (pp. 782-802).
> ###### [9] Jin, H., Li, M., Wang, X., Xu, Z., Huang, M., Jia, Y., & Lian, D. (2025, July). Internal Value Alignment in Large Language Models through Controlled Value Vector Activation. In Proceedings of the 63rd Annual Meeting of the Association for Computational Linguistics (Volume 1: Long Papers) (pp. 27347-27371).
> ###### [10] Gurnee, Wes, and Max Tegmark. "Language Models Represent Space and Time." The Twelfth International Conference on Learning Representations.
>
> ----
>
> > **Q1: How does value steering based on your discovery compare against behavioral alignment (e.g., RLHF and DPO) in aligning LLM values?**
>
> (Q1 is answered with W1)

---

### Official Review · Reviewer_DfyZ · 2025-11-01

**Soundness:** 2
**Presentation:** 3
**Contribution:** 2
**Rating:** 4
**Confidence:** 4

**Summary:**

The paper advances a dual‑pathway account of value expression in LLMs --intrinsic (without value prompts) versus prompted (with explicit value prompts)--and links their representational differences to controllability, linguistic diversity, and safety. It extracts valeu directions for ten Schwartz dimensions from residual streams via a difference‑in‑means procedure, orthogonalizes to separate shared and unique components, and uses SVD to define shared/difference neuron axes and identify shared, intrinsic‑unique, and prompted‑unique neurons. Causal interventions (vector injection, neuron amplification) are run across Qwen2.5‑7B/1.5B and Llama‑3.1‑8B in five languages while capping MMLU degradation at ≤5 points. The shared component approximates the Schwartz circular geometry and carries value semantics; prompted‑unique components deliver stronger steerability and reveal a model‑agnostic “prompted - intrinsic” compliance direction that increases adherence -- including to unsafe prompts and jailbreak rates on HarmBench/AdvBench--whereas intrinsic‑unique components increase lexical/semantic diversity and naturalness. Its contributions are: a unified, reproducible vector ‑ and neuron‑level framework with causal evidence for disentangling shared vs. unique value representations; mechanistic insight that the prompted‑unique pathway chiefly mediates instruction‑following while the intrinsic‑unique pathway drives diversity over a shared value core; and an interpretable compliance channel that enables monitoring and mitigation for alignment.

**Strengths:**

Quality. The empirical study is thorough. It spans multiple model families and scales, multiple languages, and diverse behavioral evaluations (PVQ, free-form PVQ, situational dilemmas, Value Portrait). The authors couple vector-level interventions with neuron-level amplification, which supports causal interpretation beyond correlation. The use of an MMLU-based constraint on intervention strength is a practical and responsible step toward assessing capability trade-offs. The convergence of evidence -- from geometric similarities, behavioral steering outcomes, diversity metrics, and logit-lens vocabulary projections -- creates a compelling narrative.

Clarity. The conceptual pipeline is clearly explained, the orthogonalization and SVD projections are well motivated, and the layer/strength selection procedure (with a cap on MMLU drop) is transparently described. Figures are informative and help connect geometry to behavior.

Significance. The work has immediate value for both controllable generation and safety alignment. It offers a principled way to decide when to favor prompted mechanisms (for strong steerability) versus intrinsic mechanisms (for naturalness/diversity). The identification of a general “compliance direction” suggests practical monitoring and mitigation strategies against jailbreaks. Methodologically, the shared-unique-neuron framework appears portable to other controllable attributes (e.g., politeness, emotion, persona).

**Weaknesses:**

Evaluation and labeling bias.
- Much of the labeling and evaluation rests on a single automated judge (GPT-4o-mini), with a relatively small human validation sample (n = 100 items × 3 annotators). While the reported 92.33% agreement is encouraging, it is limited in size and granularity.
- Suggestions: please expand human evaluation across languages and value dimensions, report inter-annotator agreement (e.g., κ) with confidence intervals, and include robustness checks with multiple, diverse evaluator models (including open-source judges).

Generalization of the “compliance channel.”
- The evidence for a general instruction-following direction is strong in the value domain, with positive/negative prompts and jailbreak scenarios. However, it remains unclear whether this direction broadly improves task compliance on unrelated instruction tasks (summarization, translation, extraction) without harming factuality or increasing hallucination.
- Suggestions: a small suite of atomic, non-value tasks would help establish boundary conditions and practical utility.

Linear difference-in-means and temporal information.
- The approach assumes linear separability in the residual stream after token-wise averaging, potentially discarding positional and syntactic information that could carry value cues.
- Suggestions: consider adding position-weighted or span-aligned variants of the extraction, and compare with simple linear probes or sparse regression probes to test linearity. An ablation on time-sensitive signals (e.g., clause-initial vs. clause-final tokens) would strengthen the case.

Statistical robustness of the Schwartz-circle recovery.
- The PCA visualization of shared axes is compelling, but significance versus randomized baselines is not fully quantified.
- Suggestions: please add statistical tests (e.g., Procrustes alignment) to the theoretical circle, circular correlation, or Mantel tests -- along with confidence intervals, to quantify alignment with Schwartz’s structure across layers and models.

Diversity analyses and decoding confounds.
- Although settings were held constant, distinct-n and entropy are sensitive to output length, temperature, and sampling strategy.
- Suggestions: please add controls: length-matched comparisons, temperature sweeps, and sampling-strategy ablations, together with effect sizes and confidence intervals. This would isolate representational causes from decoding artifacts.

Template and vocabulary leakage.
- Since the prompted templates explicitly assert “you value {value},” they may directly inflate value-specific keywords and exaggerate the “templated” effect attributed to prompted mechanisms.
- Suggestions: please include implicit/indirect prompts and counterfactual templates (that avoid direct value keywords but elicit similar behaviors) as controls to show that findings are not an artifact of lexical priming.

Missing related works. For example, related research using SAE features for value steering such as:
- https://www.lesswrong.com/posts/k8bBx4HcTF9iyikma/sae-features-for-refusal-and-sycophancy-steering-vectors
- https://arxiv.org/pdf/2501.00581
Maybe you could also consider comparing your method to these techniques (as well as those you mentioned in Section 5: Steering values through activation engineering). This comparison would offer a more robust way of validating your way of representing values as vectors.

**Questions:**

- The description of the value vector/neuron extraction method is vague. In lines 103-105, the calculation method for the value vector is briefly mentioned, but how can it be guaranteed that a method like "token-averaged mean difference between residual stream activations" truly yields a vector related to the intended value, and not just random differences between samples? Is there any subsequent proof for this?

- Safety trade-offs. If one suppresses the compliance direction to harden models against jailbreaks, what is the impact on desirable “policy-compliant” adherence (e.g., following benign safety instructions)? Is there measurable collateral damage to safe refusal behavior?

- Labeling robustness. Can you report cross-language, cross-value human agreement statistics with confidence intervals? How sensitive are results to the choice of evaluator model(s)? Do you observe systematic biases (e.g., over-penalizing or over-rewarding “safety” or “conformity”)?

- Statistical alignment of shared axes. Can you quantify the match to the Schwartz circle using formal statistics (e.g., circular correlation, Procrustes alignment) across layers/models and provide p-values vs. randomized baselines?

- Scope of the compliance channel. Does steering along the mean “prompted − intrinsic” direction improve adherence on standard instruction tasks (summarization, translation, extraction) without degrading faithfulness? Are there conflicts with refusal strategies on safety-related prompts?

- Decoding controls. Under length-matching and with temperature/sampling sweeps, does the intrinsic > prompted diversity advantage persist? A figure/table summarizing these robustness checks would be helpful.

---

> ### Author Response · Authors · 2025-12-03
> **Official Response to Reviewer DfyZ (Part 1/11)**
>
> We sincerely thank you for your comprehensive and constructive review. We are encouraged that you recognized the thoroughness of our empirical study across multiple model families and languages, as well as the significance of our dual-pathway framework for safety alignment and controllable generation. We particularly appreciate your positive remarks on the clarity of our conceptual pipeline and the compelling narrative supported by the convergence of geometric and behavioral evidence.
>
> We have taken your suggestions seriously and have performed additional analyses to address the concerns regarding evaluation robustness, the generalization of the compliance channel, and the comparison with related works (including SAE-based methods). Below, we provide a point-by-point response to your questions and comments.
>
> ---
>
> >**W1. Evaluation and labeling bias.**
> **Much of the labeling and evaluation rests on a single automated judge (GPT-4o-mini), with a relatively small human validation sample (n = 100 items × 3 annotators). While the reported 92.33% agreement is encouraging, it is limited in size and granularity.**
>
> > **Suggestions: please expand human evaluation across languages and value dimensions, report inter-annotator agreement (e.g., κ) with confidence intervals, and include robustness checks with multiple, diverse evaluator models (including open-source judges).**
>
>
> **A1.** Following your suggestion, we conducted human evaluation on an additional 200 items with three annotators. The agreement between gpt-4o-mini and the human annotators was 91.5%. The Fleiss’ Kappa value across the three annotators was 0.75  with a 95% confidence interval of [0.67, 0.83]. We also analyzed agreement across value dimensions. Overall, gpt-4o-mini and the annotators showed consistently high agreement across value dimensions. The below table summarizes the per-dimension agreement rates:
>
> | Value Dimension | Annotator 1 | Annotator 2 | Annotator 3 | Average |
> |-----------------|-------------|-------------|-------------|---------|
> | Self-Direction  | 80.0%       | 90.0%       | 85.0%       | 85.0%   |
> | Stimulation     | 85.0%       | 100.0%      | 80.0%       | 88.3%   |
> | Hedonism        | 75.0%       | 100.0%      | 95.0%       | 90.0%   |
> | Achievement     | 95.0%       | 90.0%       | 95.0%       | 93.3%   |
> | Power           | 95.0%       | 90.0%       | 90.0%       | 91.7%   |
> | Security        | 95.0%       | 85.0%       | 95.0%       | 91.7%   |
> | Conformity      | 95.0%       | 85.0%       | 80.0%       | 86.7%   |
> | Tradition       | 95.0%       | 95.0%       | 90.0%       | 93.3%   |
> | Benevolence     | 95.0%       | 90.0%       | 100.0%      | 95.0%   |
> | Universalism    | 100.0%      | 95.0%       | 100.0%      | 98.3%   |
>
> In addition, following your suggestion to evaluate with diverse evaluator models, including open-source judges, we repeated the Situational Dilemmas experiment (Section 3.2.2, Table 5) using the following models as judges: Qwen2.5-72B-Instruct, Qwen3-Next-80B-A3B-Instruct, and gpt-4.1-mini. We further analyzed the inter-model agreement between these judges and our original judge model gpt-4o-mini. The Fleiss’ Kappa values were 0.44 for intrinsic setting, and 0.43 for the prompted setting, showing moderate agreement across evaluator models. We believe this level of agreement is partly due to the three-category evaluation protocol (win, lose, tie), which tends to introduce more disagreement across judge models compared to binary categorization (win, lose). In conclusion, gpt-4o-mini is a reliable annotator that shows high agreement with humans, which justifies our use of it.

---

> ### Author Response · Authors · 2025-12-03
> **(Part 2/11)**
>
> > **W2. Generalization of the “compliance channel.”**
> **The evidence for a general instruction-following direction is strong in the value domain, with positive/negative prompts and jailbreak scenarios. However, it remains unclear whether this direction broadly improves task compliance on unrelated instruction tasks (summarization, translation, extraction) without harming factuality or increasing hallucination.**
>
> >**Suggestions: a small suite of atomic, non-value tasks would help establish boundary conditions and practical utility.**
>
>
> **A2.**
>
> We agree that it is important to understand how far the “compliance channel” extends to unrelated instruction tasks. Following your suggestions, we conduct new experiments and show that the prompted-unique component enhances instruction-following in
> In response, we designed two complementary non‑value evaluations:
>
> (1) a non-value **translation** task where compliance is defined as following an explicit gender instruction, and
>
>
> (2) a suite of **atomic, content-neutral constraints**.
>
> Across these settings, we find that the compliance channel can (1) improve instruction satisfaction when the target behavior is already within the model’s capabilities , but (2) does not act as a universal booster for low-level constraint satisfaction. We therefore position it as a targeted mechanism for modulating behavior in line with specified guidance, not as a general-purpose compliance direction.
>
> ### **(1) Translation with explicit gender instructions.**
>
> To directly test “task compliance” on a non-value task, we use the GAMBIT gender translation dataset [1]. Each example is an English sentence where the grammatical gender of the target referent is ambiguous, and the model is asked to translate it into Spanish or French with an explicit gender instruction.
> Concretely, prompts contain instructions such as
> > “… translate so that the [profession] is referred to as a woman.”
> or
> >”… translate so that the [profession] is referred to as a man.”
>
> We measure *gender-instruction accuracy*: the fraction of translations where the detected grammatical gender in the output matches the requested label, counting translations with no detectable gendered pronoun as failures. We apply the compliance-channel steering (same extraction procedure as in Section~4.3) to Qwen-2.5-7B-Instruct and Llama-3.1-8B-Instruct, varying the steering weight $w$. The overall gender-instruction accuracy behaves as follows:
>
> | Model | Steering weight (w) | Gender accuracy |
> |--------------------------|----------------------:|----------------:|
> | Qwen‑2.5‑7B‑Instruct | 0 (no steering) | 0.40 |
> | | 4 | 0.41 |
> | | 8 | **0.45** |
> | Llama‑3.1‑8B‑Instruct | 0 (no steering) | 0.46 |
> | | 4 | 0.45 |
> | | 8 | **0.52** |
>
> Thus, on this non-value translation task, steering mostly **increases the rate at which the model follows the explicit gender instruction** in the prompt. This directly addresses your notion of “task compliance on unrelated instruction tasks”.
>
>
> ### **(2) Atomic, content-neutral instruction following.**
>
> Next, we follow your suggestion of “a small suite of atomic, non-value tasks”, and also evaluate the compliance channel on IFEVAL [2]. IFEVAL consists of automatically checkable instructions over simple tasks (e.g., “include the word X exactly twice”, “answer in JSON with keys A,B”, length and punctuation constraints) with no explicit value content. Here, *task compliance* is simply the fraction of generations that satisfy the given constraint. The results are shown in the table below:
>
> | Model                      | Steering weight | Keywords | Length | Detectable format | Punctuation | Overall accuracy |
> |----------------------------|-------------:|---------:|-------:|------------------:|------------:|----------------:|
> | Llama3.1-8b-instruct       | 0 (base)     | **0.233** | **0.364** | **0.529** | **0.818** | **0.303** |
> |                            | 4            | 0.160    | 0.273  | 0.344             | 0.530       | 0.152       |
> |                            | 8            | 0.160    | 0.273  | 0.369             | 0.379       | 0.146       |
> | Qwen2.5-1.5b-instruct      | 0 (base)     | **0.166** | **0.301** | **0.446** | **0.212**       | **0.209** |
> |                            | 4            | 0.110    | 0.238  | 0.159             | 0.258       | 0.092       |
> |                            | 8            | 0.129    | 0.231  | 0.121             | 0.348   | 0.096       |
> | Qwen2.5-7b-instruct        | 0 (base)     | 0.200    | 0.330  | **0.470**         | **0.230**   | 0.285       |
> |                            | 4            | 0.213    | **0.350** | 0.440         | 0.214       | 0.292       |
> |                            | 8            | **0.232** | 0.347  | 0.462             | 0.205       | **0.304** |

---

> ### Author Response · Authors · 2025-12-03
> **(Part 3/11)**
>
> *(continued)*
>
> For Llama-3.1-8B and Qwen-2.5-1.5B, steering along the compliance channel actually reduces IFEVAL instruction-following, roughly halving overall accuracy (e.g. 0.303 → 0.15 for Llama-3.1-8B-Instruct) with similar drops across constraint types. Qwen-2.5-7B-Instruct shows only a modest overall gain, mainly on keyword and length constraints, which is consistent with our earlier finding that prompted-only components emphasize prompt-related tokens and shorten responses. Taken together, these results suggest a natural boundary condition: the compliance channel operates within the model’s existing capability region rather than creating new low-level skills. When a model already struggles with atomic constraints (e.g. Qwen-2.5-1.5B at 0.209 overall accuracy), steering cannot make it reliably satisfy them; it mainly redistributes behavior rather than upgrading core constraint-following ability.
>
> Taken together, these experiments clarify how the compliance channel affects task compliance beyond our original value-evaluation setup. On a realistic, non-value translation task with explicit gender instructions (GAMBIT), it improves instruction satisfaction when the underlying capability (translation and gender marking) is already present. On atomic, content-neutral constraints (IFEVAL), reusing the same direction and weights does not yield a universal boost and can even reduce constraint satisfaction when base performance is modest, suggesting that the channel modulates how existing behaviors are expressed rather than creating new low-level skills. Accordingly, we now present the compliance channel as a targeted mechanism for steering behavior in line with explicit guidance, and explicitly avoid claiming general gains in factuality or hallucination on broad summarization/translation/extraction benchmarks, which we leave for future work.
>
>
> ###### [1] Mastromichalakis, O. M., Filandrianos, G., Symeonaki, M., & Stamou, G. Assumed Identities: Quantifying Gender Bias in Machine Translation of Gender-Ambiguous Occupational Terms. In Greeks in AI Symposium 2025.
>
>
> ###### [2] Zhou, J., Lu, T., Mishra, S., Brahma, S., Basu, S., Luan, Y., ... & Hou, L. (2023). Instruction-following evaluation for large language models. arXiv preprint arXiv:2311.07911.
>
>
>
> ---
>
> >**W3. Linear difference-in-means and temporal information.**
> **The approach assumes linear separability in the residual stream after token-wise averaging, potentially discarding positional and syntactic information that could carry value cues.**
>
> >**Suggestions: consider adding position-weighted or span-aligned variants of the extraction, and compare with simple linear probes or sparse regression probes to test linearity. An ablation on time-sensitive signals (e.g., clause-initial vs. clause-final tokens) would strengthen the case.**
>
>
> **A3.**
> Thank you for raising this concern about positional and temporal information. Following your suggestions, we added (i) span-based positional ablations and (ii) a comparison between difference-in-means vectors and linear probes trained on the same residual activations. In summary, we find that (a) value information is linearly decodable from token-averaged residual activations, including when restricted to specific spans, but (b) global token-averaged difference-in-means vectors consistently yield the strongest and most stable steering, while probe-based directions often have weaker and sometimes negative effects on value scores. These results are now included in the revised manuscript (Appendix A.5) and we summarize them below.
>
> ### **Span-based ablations and their implications.**
>
> Following the suggestion on span-aligned variants, we recomputed value vectors on a 10% subsample of the situational-dilemma extraction set using four windows at a fixed steering layer:
>
> (i) all tokens in the response (our original method),
> (ii) the first 5 tokens,
> (iii) the middle 5 tokens, and
> (iv) the final 5 tokens.
>
> For each value and window, we evaluated steering on the multilingual PVQ dataset by sweeping the steering strength and measuring the change in mean value score (1–6 scale, averaged over the ten values) relative to the unsteered baseline.

---

> ### Author Response · Authors · 2025-12-03
> **(Part 4/11)**
>
> *(continued)*
>
> | Model         | Weight | All tokens         | First 5             | Middle 5            | Final 5             |
> |--------------|--------|-------------------:|--------------------:|--------------------:|--------------------:|
> | Qwen2.5-7B   | 0      | 2.92               | 2.92                | 2.92                | 2.92                |
> |   | 5 (intrinsic)    | **5.35 (+2.43)** | 5.02 (+2.10)    | 3.05 (+0.13)        | 3.88 (+0.96)|
> |   | 5 (prompted)    | **5.76 (+2.84)**   | 5.20 (+2.29)*   | 3.73 (+0.81)   | 4.21 (+1.29)    |
> | Llama-3.1-8B | 0      | 3.26               | 3.26            | 3.26            | 3.26            |
> | | 5 (intrinsic)       | **4.68 (+1.42)**   | 3.14 (−0.12)        | 1.85 (−1.41)        | 1.92 (−1.34)        |
> | | 5 (prompted)        | **5.12 (+1.88)**   | 3.89 (+0.62)        | 1.50 (−1.76)        | 2.07 (−1.20)        |
>
> At a representative steering weight $w = 5$, we observe that for both models, the vectors computed using all tokens yield the largest increase in mean PVQ score, while the first-, middle-, and final-5-token vectors produce noticeably smaller gains and can even decrease the target value score.
>
> These span-based ablations address the reviewer’s concern that token-wise averaging might discard temporal or syntactic cues. Restricting extraction to clause-initial or clause-final slices still changes PVQ scores, which indicates that early, middle, and late parts of the answer each carry some value-related signal. However, the fact that these span-based vectors consistently weaken steering and sometimes flip its direction suggests that no single span suffices to capture the full value mechanism.
>
> The fact that these position-controlled vectors still underperform the all-token vector therefore suggests that value-related features are distributed across positions and that, for our goal of capturing a global value mechanism, aggregating evidence from all tokens before computing the class-wise mean provides a more reliable causal steering direction than restricting the extraction to any particular span.
>
> ### **Linear probes and explicit tests of linearity**
>
> To examine linearity more directly, we followed the suggestion to compare difference-in-means vectors with linear probes trained on the same representations. For each of the four windows (all, first 5, middle 5, final 5), we trained a logistic-regression probe to predict whether the target value is expressed (value expressed vs. value unexpressed) from the token-averaged residual stream at the steering layer.
>
> On a held-out test set, these probes achieve high predictive performance (F1 ≈ 0.95), indicating that value expression is linearly decodable from the token-averaged residual stream, even when restricted to a single span. This directly addresses the reviewer’s concern about the linear separability assumption: simple linear probes on the same averaged activations achieve strong classification, so the labels are indeed approximately linearly separable in this space. However, when we re-use the probe weight vectors themselves as steering directions, their causal effect is weaker and less stable than the difference-in-means direction. At $w = 5$, steering results are shown in the following table:
>
> | Model         | Method            | All tokens | First 5  | Middle 5 | Final 5  |
> |--------------|-------------------|-----------:|---------:|---------:|---------:|
> | Qwen2.5-7B   | difference-in-means (intrinsic, w = 5)   | **+2.41**      | **+2.55**    | +0.35    | **+1.62**    |
> |  | logistic regression (intrinsic, w = 5)   | +1.72      | +1.15    | **+1.28**    | −0.55    |
> | Qwen2.5-7B   | difference-in-means (prompted, w = 5)   | **+2.84**      | **+2.29**    | +0.81    | **+1.95**    |
> |  | logistic regression (prompted, w = 5)   | +1.87      | +1.76    | **+1.10**    | −0.18    |
> | Llama-3.1-8B | difference-in-means (intrinsic, w = 5)   | **+1.45**      | **−0.12**    | −1.35    | −1.25    |
> | | logistic regression (intrinsic, w = 5)   | +0.88      | +0.65    | **−0.34**    | **−1.12**    |
> | Llama-3.1-8B | difference-in-means (prompted, w = 5)   | **+1.88**      | **+0.62**    | −1.76    | −1.20    |
> | | logistic regression (prompted, w = 5)   | +1.24      | +0.51    | **+0.13**    | **−0.67**    |
>
> We find that difference-in-means based steering produces larger and more consistent changes in value scores across models and languages, while linear probe based directions often have weaker effects and can even reduce the target value score for some cases.

---

> ### Author Response · Authors · 2025-12-03
> **(Part 5/11)**
>
> *(continued)*
>
> We believe this gap reflects the geometry of high-dimensional representations. For a given value, multiple separating hyperplanes may achieve high classification accuracy. Logistic regression tends to select the hyperplane that maximizes accuracy, but high accuracy alone does not necessarily indicate that the direction is closely related to the underlying value mechanism. In contrast, the difference-in-means vector explicitly points from the centroid of unexpressed activations to the centroid of expressed activations, providing a single canonical direction that reflects how the representation shifts on average when the value is present. Empirically, this centroid-based direction produces larger and more monotonic changes in value scores across models, languages, and tasks, compared to linear-probe directions trained on the same features.
>
> ### **Clarifying our rationale for token-averaged difference-in-means vectors.**
>
> These new experiments also help clarify why we use token-averaged difference-in-means vectors in the first place. Our goal in value vector extraction(Section 2.1) is to isolate global value mechanisms that determine the stance of an entire answer, rather than local lexical patterns tied to specific spans. We therefore treat value expression as a sentence-/discourse-level property and aggregate residual-stream activations over all tokens in the response before computing the mean difference between value expressed and value unexpressed sets. The steering experiments in the main paper already show that adding a scalar multiple of this token-averaged difference-in-means vector at a fixed layer produces stable, approximately monotonic changes in value expression as we increase the steering strength, across four languages and three evaluation settings (PVQ questionnaires, situational dilemmas, and Value Portrait) (Figure 4, Table 1, Figure 5, Figure 6).
>
> The span-based ablations and linear-probe comparisons now provide additional evidence that value-relevant information is spread across different positions, so restricting to a single span tends to remove important signals and degrades steering performance. Additionally, we also showed that difference-in-means direction defined on the token-averaged activations yields stronger and more consistent causal interventions than linear-probe directions trained on the same features. Taken together, these findings support our choice of token-averaged difference-in-means vectors as a compact, position-aggregated representation of value mechanisms that still preserves the key linear structure needed for effective steering. **We have incorporated the new span-based and probe-based analyses into the revised manuscript (Appendix A.5) and clarified this rationale in Section 2.1.**

---

> ### Author Response · Authors · 2025-12-03
> **(Part 6/11)**
>
> >**W4. Statistical robustness of the Schwartz-circle recovery.**
> **The PCA visualization of shared axes is compelling, but significance versus randomized baselines is not fully quantified.**
>
> >**Suggestions: please add statistical tests (e.g., Procrustes alignment) to the theoretical circle, circular correlation, or Mantel tests -- along with confidence intervals, to quantify alignment with Schwartz’s structure across layers and models.**
>
> **A4.**
>
> We agree that the PCA visualization of the shared value axes (Figure 7) is mainly descriptive and does not by itself quantify statistical robustness. Therefore, in the revision we added orthogonal Procrustes alignment tests between the learned shared axes and Schwartz’s theoretical circumplex. As in our steering experiments, we focus on the mid layer (e.g. layer 14 for Qwen2.5‑7B‑Instruct) for this analysis.
>
>
> Specifically, we perform an orthogonal Procrustes analysis, which finds the best‑fitting rotation (and uniform scaling) between two point sets and summarizes goodness of fit with a “disparity” term. We report $R^2 =1−\text{disparity}$ as an alignment score between 0 and 1. We consider two closely related variants:
>
>
> 1. **Four higher-order domains.**
>    Since the ten values of Schwartz’s theory are often grouped and analyzed as the four higher order values (Openness to Change, Conservation, Self-Transcendence, and Self-Enhancement), we first group the ten shared axes into these four higher‑order values and average the shared directions within each value. For each model, we take the 2D embedding of these four higher-order‑value-level shared axes (PCA at layer 14) and align this 4‑point configuration to a canonical Schwartz circle with four equally spaced value positions. As baselines, we compare with (a) random orthonormal directions in the same dimensionality and (b) random permutations of the value labels applied to the shared axes. The resulting Procrustes $R^2$ values and 95% confidence intervals are:
>
>    | Model                 | Shared axes (95% CI)        | Random directions (95% CI)     | Shared axes + permuted labels (95% CI) |
>    |-----------------------|------------------------------|----------------------------------|----------------------------------------|
>    | Qwen2.5-7B-Instruct   | **0.707 (0.664–0.750)**      | 0.558 (0.441–0.675)             | 0.556 (0.457–0.650)                    |
>    | Qwen2.5-1.5B-Instruct | **0.595 (0.435–0.721)**      | 0.532 (0.391–0.673)             | 0.464 (0.297–0.631)                    |
>    | Llama3.1-8B-Instruct  | **0.643 (0.495–0.831)**      | 0.532 (0.389–0.675)             | 0.472 (0.353–0.591)                    |
>
>    The shared axes show relatively high alignment scores ($R^2 \approx 0.6\text{-}0.7$) and, for all three models, the point estimates lie above both the random‑direction and permuted‑label baselines. This suggests that the learned shared components recover the expected circular arrangement of the four superordinate Schwartz domains, which is arguably the most theoretically salient level of the framework.
>
>
>
>
>
>
> 2. **Ten fine-grained values.**
>    We then apply the same Procrustes analysis at the finer level of all ten Schwartz values: for each model, we PCA‑embed the ten shared axes in 2D (as in Figure 7), align them to the canonical ten‑point Schwartz circle, and compare the resulting Procrustes ($R^2$) to baselines from (a) random orthonormal directions and (b) permutations of the value labels. The table below reports these ($R^2$) scores and their 95% confidence intervals.
>
>
>    | Model                    | Shared axes (95% CI)        | Random directions (95% CI)     | Shared axes + permuted labels (95% CI) |
>    |--------------------------|------------------------------|----------------------------------|----------------------------------------|
>    | Qwen2.5-7B-Instruct      | **0.294 (0.281–0.309)**      | 0.231 (0.220–0.243)             | 0.170 (0.160–0.181)                    |
>    | Qwen2.5-1.5B-Instruct    | **0.244 (0.230–0.258)**      | 0.239 (0.226–0.249)             | 0.167 (0.155–0.178)                    |
>    | Llama3.1-8B-Instruct     | **0.254 (0.241–0.268)**      | 0.247 (0.234–0.259)             | 0.165 (0.154–0.177)                    |
>
>    Alignment scores at the ten‑value level are lower than at the higher-order-value level, as the finer-grained value positions contain more noise and model‑specific idiosyncrasies than the broader higher-order value boundaries. Nevertheless, across all three models the shared‑axis alignment remains clearly higher than the permuted‑label baseline and comparable to or slightly better than the random‑direction baseline, especially for the largest model (Qwen2.5‑7B‑Instruct). In other words, even at the more fine‑grained level, the shared axes preserve more of the Schwartz structure than we would expect from chance alone.

---

> ### Author Response · Authors · 2025-12-03
> **(Part 7/11)**
>
> *(continued)*
>
> For each model and variant, we estimate 95% confidence intervals with bootstrap resampling over the extraction set and obtain empirical p values against both versions of the baseline. Taken together, these analyses show that (i) at the level of the four higher‑order Schwartz values, the shared components form a configuration that strongly resembles the theoretical circle, and (ii) at the level of all ten values the alignment is weaker but still reliably above label‑permutation baselines. Overall, this indicates that the recovered shared value axes are not merely visually reminiscent of the Schwartz circle but exhibit a reproducible geometric alignment with its structure across layers and models. **We have incorporated these results into Section 4.1 of the revised manuscript.**
>
>
>
>
>
>
> >**W5. Diversity analyses and decoding confounds.**
>
> >**Although settings were held constant, distinct-n and entropy are sensitive to output length, temperature, and sampling strategy.**
>
> >**Suggestions: please add controls: length-matched comparisons, temperature sweeps, and sampling-strategy ablations, together with effect sizes and confidence intervals. This would isolate representational causes from decoding artifacts.**
>
>
> **A5.**
>
> We agree that Distinct-n and entropy can be influenced by decoding choices, and we have updated our analyses to better separate decoding artifacts from representational effects.  In the revised manuscript (Section 3.3 and Appendix G), we (i) add a length-robust diversity metric and (ii) run temperature and sampling sweeps, which together show that intrinsic generations remain more diverse than prompted generations across metrics and decoding settings Below we summarize these controls and how they affect our interpretation of the diversity results.
>
> ###  **Length controls with expectation-adjusted Distinct (EAD)**
>
> Instead of relying only on raw Distinct-n, which tends to decrease as generations become longer [3], we now also report Expectation-Adjusted Distinct (EAD) [4], which normalizes Distinct-n by its expected value under a length-matched random baseline.
>
> For each model and setting we recompute our main situational-dilemma diversity table with EAD-2/3 in addition to Distinct-2/3 and entropy. We report the table for Qwen2.5-7B-Instruct as below:
>
>
> | Setting              | Distinct-2 / 3 ↑ | **EAD-2 / 3 ↑**      | Entropy-2 / 3 ↑ | Embedding variation ↑ | Frequently occurring words (Achievement) |
> |----------------------|------------------|----------------------|------------------|------------------------|------------------------------------------|
> | Intrinsic            | 0.362 / 0.654    | **0.298 / 0.552**    | 12.743 / 14.361  | 0.563                  | work, project, high                      |
> | Prompted             | 0.342 / 0.619    | **0.298 / 0.547**    | 12.191 / 13.790  | 0.549                  | achievement, growth, goals               |
> | Intrinsic_Orthogonal | 0.402 / 0.713    | **0.345 / 0.627**    | 13.130 / 14.735  | 0.568                  | provide, consider, term                  |
> | Prompted_Orthogonal  | 0.203 / 0.343    | **0.182 / 0.312**    | 12.459 / 13.907  | 0.555                  | achieve, excellence, goal                |
>
> The ordering reported in the paper is preserved and becomes slightly clearer when using EAD-n: intrinsic steering yields higher adjusted diversity than prompted steering, and the intrinsic-orthogonal component achieves the highest EAD-2/3, whereas the prompted-orthogonal component is the lowest. This indicates that our main diversity effect is not an artifact of average response length.
>
>
> ### **Temperature and sampling sweeps.**
> Following your suggestion, we have also evaluated diversity under a range of decoding hyperparameters, using the same situational-dilemma dataset and evaluation protocol as in the main text.
>
> - **Temperature sweeps (fixed top-p).**
> We fix top-p = 1.0 and vary temperature $T \in \\{0.3, 0.7, 1.0\\}$. For each value of T, we compute mean EAD-2/3, 2-gram/3-gram entropies, and embedding-space variance for both intrinsic and prompted modes. The following table shows the results on Qwen2.5-7b-Instruct:
>
>
>    |   T   | Mode      | EAD-2 / 3 ↑      | Entropy-2 / 3 ↑    | ‖σ²‖ mean ↑ |
>    |-------|-----------|------------------|--------------------|------------:|
>    | 0.3   | Intrinsic | 0.370 / 0.641    | **8.575 / 9.707**      | **0.01397**     |
>    |       | Prompted  | **0.372 / 0.645**    | 8.525 / 9.628      | 0.01390     |
>    | 0.7   | Intrinsic | **0.397 / 0.679**    | **8.689 / 9.819**      | **0.01399**     |
>    |       | Prompted  | 0.395 / 0.676    | 8.640 / 9.737      | 0.01397     |
>    | 1.0   | Intrinsic | 0.432 / 0.718    | **8.851 / 9.955**      | 0.01403     |
>    |       | Prompted  | **0.434 / 0.722**    | 8.809 / 9.884      | 0.01403     |

---

> ### Author Response · Authors · 2025-12-03
> **(Part 8/11)**
>
> *(continued)*
>
> **Top-p sweeps (fixed temperature).**
> For the top-p sweeps, we fix temperature $T = 0.7$ and sweep top-p $ \in \\{1.0, 0.9, 0.7, 0.5\\}$. The following table also shows the results on Qwen2.5-7b-Instruct:
>
> |   p   | Mode      | EAD-2 / 3 ↑      | Entropy-2 / 3 ↑    | ‖σ²‖ mean ↑ |
>   |-------|-----------|------------------|--------------------|------------:|
>   | 1.0   | Intrinsic | **0.397 / 0.679**    | **8.697 / 9.825**      | **0.01399**     |
>   |       | Prompted  | 0.395 / 0.676    | 8.637 / 9.735      | 0.01397    |
>   | 0.9   | Intrinsic | 0.383 / 0.660    | **8.643 / 9.777**      | **0.01398**     |
>   |       | Prompted  | **0.385 / 0.665**    | 8.588 / 9.690      | 0.01391     |
>   | 0.7   | Intrinsic | 0.373 / 0.644    | **8.589 / 9.722**      | **0.01399**     |
>   |       | Prompted  | **0.375 / 0.648**    | 8.539 / 9.640      | 0.01391     |
>   | 0.5   | Intrinsic | 0.368 / 0.636    | **8.561 / 9.691**     | **0.01399**     |
>   |       | Prompted  | **0.371 / 0.640**    | 8.518 / 9.616      | 0.01388     |
>
>
> Across temperature and top-p sweeps, we observe a consistent pattern. Intrinsic generations match or exceed prompted generations on the majority of diversity metrics (n-gram entropy and embedding-variance norms) at every tested T and p, while EAD-2/3 remains closely matched between modes and only shows small fluctuations in either direction. Changing decoding hyperparameters mainly rescales overall diversity (e.g., higher entropy at larger T) but never reverses the intrinsic–prompted ordering on most metrics. These results indicate that the observed diversity gap remains consistent across different temperatures, sampling strategies, and length distributions.
>
>
> ###### [3] Li, J., Galley, M., Brockett, C., Gao, J., & Dolan, B. (2016). A diversity-promoting objective function for neural conversation models. In Proceedings of the 2016 Conference of the North American Chapter of the Association for Computational Linguistics: Human Language Technologies (NAACL-HLT).
>
> ###### [4] Liu, S., Sabour, S., Zheng, Y., Ke, P., Zhu, X., & Huang, M. (2022). Rethinking and refining the Distinct metric. In Proceedings of the 60th Annual Meeting of the Association for Computational Linguistics (ACL 2022), Short Papers.
>
>
>
> >**W6. Template and vocabulary leakage.**
> **Since the prompted templates explicitly assert “you value {value},” they may directly inflate value-specific keywords and exaggerate the “templated” effect attributed to prompted mechanisms.**
>
> >**Suggestions: please include implicit/indirect prompts and counterfactual templates (that avoid direct value keywords but elicit similar behaviors) as controls to show that findings are not an artifact of lexical priming.**
>
>
> **A6.**
>
> We understand the concern that our original prompted templates (“you value {value} …”) might introduce lexical priming by directly inserting value labels and salient keywords (e.g., success for Achievement), and that this could exaggerate the “templated” effect. In the revision we therefore add a set of implicit, keyword-free prompts and extract a set of “implicitly” prompted value directions.
>
> We introduce implicit counterfactual prompts that express each value only through behavioral descriptions, without ever naming the value or using its typical keywords. Specifically, we use the items from PVQ-40, and rewrite them into imperative, behavior-focused descriptions of how a person with a certain value dimension tends to act.
>
>
> Below  is an example of the revised system prompts.
>
> - Original system prompt (Achievement):
> > *“You are a person who strongly values **achievement**. You care about success, ambition, and standing out from others by reaching demanding goals.”*
>
> - Implicit system prompt (Achievement):
>  > *“You tend to seek out difficult tasks, set demanding objectives for yourself, and feel most satisfied when your efforts lead to challenging accomplishments that others recognize as impressive.”*
>
> These implicit templates preserve the behavioral instructions but do not mention the value label or its canonical keywords. We then extract  value vectors from these implicit prompts using the same pipeline as for the explicit prompts (see Section 2.1).
>
> We now compare the diversity of text generated while steered with these implicitly‑prompted vectors.  For Qwen‑2.5‑7B (English), using the same metric definitions but over the full situational‑question dataset, the results are shown as below:
>
>
>
> | Setting      | Distinct‑2 / 3 ↑ | **EAD‑2 / 3 ↑** | Entropy‑2 / 3 ↑ | Embedding variance ‖σ²‖ ↑ |
> |-------------|-----------:|------------------|-----------------|-----------------:|
> | Intrinsic   | 0.346 / **0.637**    | **0.422 / 0.682**   |**8.682 / 9.816**    | **0.036**                      |
> | Prompted (original) | 0.280 / 0.531    | 0.322 / 0.556   | 8.163 / 9.306    | 0.034                      |
> | *Implicitly Prompted*      | **0.348** / 0.626| 0.400 / 0.655| 8.478 / 9.574 | 0.035 |

---

> ### Author Response · Authors · 2025-12-03
> **(Part 9/11)**
>
> *(continued)*
>
> Even when we remove all explicit value labels and high-salience synonyms from the system prompts, prompted generations still show reduced diversity relative to intrinsic steering (especially in EAD and entropy). The reduced gap between the intrinsic steering and the implicitly-prompted steering (compared to the original prompted steering) is expected. By replacing the explicit value label and its canonical synonyms with broader behavioral descriptions, the implicit templates yield prompted vectors that are less lexically concentrated, so prompted generations become somewhat more diverse even though they still remain less diverse than intrinsic steering on most metrics. This suggests that the intrinsic–prompted diversity gap is not solely an artifact of lexical priming from the phrase “you value {value}” and related keywords.
>
> ### **Vocabulary projection analysis.**
>
> We also repeat our logit-space vocabulary projection (see Section 4.2) for the implicit vectors and measure mean normalized entropy over the induced vocabulary distributions. The following table shows the results for Qwen2.5-7B-Instruct:
>
> | Vector type          | Mean normalized entropy |
> |----------------------|-------------------------|
> | Intrinsic            | **0.313**               |
> | Prompted             | 0.141           |
> | Implicitly Prompted  | 0.159            |
>
> The results show that implicitly prompted vectors are still more lexically concentrated than intrinsic vectors, albeit less concentrated than the original prompted ones.
>
> Together with the behavioral diversity results above, these controls indicate that while lexical priming amplifies the templated effect, the reduced diversity of prompted generations primarily reflects differences in the underlying value mechanisms rather than direct leakage of value labels or vocabulary from the prompt.
>
> > **W7. Missing related works. For example, related research using SAE features for value steering such as:
> https://www.lesswrong.com/posts/k8bBx4HcTF9iyikma/sae-features-for-refusal-and-sycophancy-steering-vectors
> https://arxiv.org/pdf/2501.00581 Maybe you could also consider comparing your method to these techniques (as well as those you mentioned in Section 5: Steering values through activation engineering). This comparison would offer a more robust way of validating your way of representing values as vectors**
>
> **A7.**
> Thank you for pointing this out. In the revised manuscript, we have expanded Section 5 and the related work section to explicitly discuss SAE-based value and refusal steering, as well as closely related activation-steering methods. Specifically, we now add a short paragraph situating our dense value vectors alongside: SAE features for refusal and sycophancy steering [5, 6], SAE-based steering of Schwartz-style values with causal-graph structure [7], sparse-space activation steering for alignment-relevant behaviors [8], SAE-denoised concept vectors that refine standard steering directions [9], and dense activation steering for instruction following [10]. This provides a clearer conceptual connection between our approach and existing SAE-based feature-steering techniques.
>
> In addition, we have added new neuron-level analyses that are closely aligned with the goals of SAE-based feature discovery. In Sections 4.1–4.2 and Appendix K, we analyze the individual neurons that contribute most to the 'shared', 'unique-intrinsic', and 'unique-prompted' value subspaces identified by our SVD-based decomposition. Following recent work on automated neuron explanations [11, 12], we collect top-activating contexts for each such neuron and obtain natural-language explanations from a neuron-explainer model. We find that shared neurons are consistently described as encoding abstract, high-level value semantics (e.g. religious rituals and heritage for Tradition, institutional safety and risk for Security, community well-being for Benevolence), prompted-unique neurons focus on explicit value definitions and value-related keywords, and intrinsic-unique neurons respond to contextual cues and scenarios that tend to co-occur with the value in natural language. This shows that the directions we extract decompose into interpretable, value-related units at the neuron level, in a way that is similar in spirit to the human-interpretable sparse features exposed by SAEs. We also report neuron amplification experiments that directly test the causal influence of these neuron groups (see Section 3.2, Table 8, and Figure 25) which is conceptually analogous to steering with selected SAE features. We show that by selectively amplifying the neurons most aligned with the shared and unique subspaces, we can steer value expression on PVQ and situational dilemmas benchmarks.
>
> Taken together, these vector-level and neuron-level results complement the SAE-based literature by showing that our value representations are not only linearly decodable but also causally effective at the neuron level.

---

> ### Author Response · Authors · 2025-12-03
> **(Part 10/11)**
>
> *(continued)*
>
> ###### [5] Shabalin, S., Kharlapenko, D., Conmy, A., & Nanda, N. (2024, October 12). SAE features for refusal and sycophancy steering vectors. AI Alignment Forum. https://www.alignmentforum.org/posts/k8bBx4HcTF9iyikma/sae-features-for-refusal-and-sycophancy-steering-vectors
> ###### [6] O’Brien, K., Majercak, D., Fernandes, X., Edgar, R., Bullwinkel, B., Chen, J., Nori, H., Carignan, D., Horvitz, E., & Poursabzi-Sangde, F. (2024). Steering language model refusal with sparse autoencoders. arXiv. https://arxiv.org/abs/2411.11296
> ###### [7] Kang, Y., Wang, J., Li, Y., Zhong, F., Feng, X., Wang, M., Tu, W., Wang, Q., Li, H., & Zheng, Z. (2025). Causal graph guided steering of LLM values via prompts and sparse autoencoders. arXiv. https://arxiv.org/abs/2501.00581
> ###### [8] Bayat, R., Rahimi-Kalahroudi, A., Pezeshki, M., Chandar, S., & Vincent, P. (2025). Steering large language model activations in sparse spaces. arXiv. https://arxiv.org/abs/2503.00177
> ###### [9] Zhao, H., Wu, X., Yang, F., Shen, B., Liu, N., & Du, M. (2025). Denoising concept vectors with sparse autoencoders for improved language model steering. arXiv. https://arxiv.org/abs/2505.15038
> ###### [10] Stolfo, A., Balachandran, V., Yousefi, S., Horvitz, E., & Nushi, B. (2025). Improving instruction-following in language models through activation steering. In Proceedings of the International Conference on Learning Representations (ICLR 2025).
>
> ###### [11] https://openaipublic.blob.core.windows.net/neuron-explainer/paper/index.html
>
>
> ###### [12] The Importance of Prompt Tuning for Automated Neuron Explanations NIPSW
>
>
> > **Q1.** The description of the value vector/neuron extraction method is vague. In lines 103-105, the calculation method for the value vector is briefly mentioned, but how can it be guaranteed that a method like "token-averaged mean difference between residual stream activations" truly yields a vector related to the intended value, and not just random differences between samples? Is there any subsequent proof for this?
>
>
> **A8.**
>
> The effectiveness of these vectors in value steering, as demonstrated in our extensive experiments, clearly shows that these vectors capture value-related signals, even across languages, rather than simply capturing "random" differences between samples. Below, we explain the theoretical justification of using the difference-in-means approach in our task, along with empirical evidence that these vectors capture value-related signals.
>
> On the theoretical side, recent work on difference-in-means concept editing provides a justification for using this direction. These results show that if the target concept is at least weakly linearly decodable (i.e., some linear probe can perform better than a constant baseline), then any linear direction that predicts this concept must be non-trivially aligned with the difference-in-means direction. Moreover, among interventions that edit activations by adding a single fixed vector, moving along this difference-in-means direction yields the largest guaranteed effect on the underlying linear concept [13]. Complementary empirical work on mass-mean probing [14] shows that difference-in-means directions match or slightly outperform logistic-regression probes on true/false classification, and are more causally effective when used for activation steering. Motivated by this theory and evidence, we adopt the same estimator.
>
> In our paper, we explicitly verify that the resulting directions are not random in several ways. First, our steering experiments show that these directions consistently and predictably shift value scores across multiple benchmarks and languages (Section 3.1), which would be very unlikely if we were steering along random differences between the samples used for vector extraction.
>
> Second, when we analyze the ten value directions jointly, we find that their shared components recover the well-known circular geometry of Schwartz's value theory, and that their highest-contributing neurons receive coherent value-related interpretations in our neuron analysis (Section 4.1 and Appendix K). This structure is difficult to explain if the vectors merely capture arbitrary differences between two sets of responses.
>
> Taken together, these results suggest that our vectors capture structure that is systematically tied to value expression. We do not, however, claim to identify a **unique ground-truth** value neuron or a mathematically optimal direction. Instead, we use the difference-in-means vector as a theoretically motivated and empirically validated value-related feature, supported by both prior work and our stability and steering analyses, rather than as an arbitrary difference between samples. **We clarified this distinction in the revised manuscript (Appendix A).**

---

> ### Author Response · Authors · 2025-12-03
> **(Part 11/11)**
>
> *(continued)*
>
> ###### [13] Nora Belrose. Diff-in-means concept editing is worst-case optimal: Explaining a result by Sam Marks and Max Tegmark. https://blog.eleuther.ai/diff-in-means, December 2023.
> ###### [14] Marks, S., & Tegmark, M. The Geometry of Truth: Emergent Linear Structure in Large Language Model Representations of True/False Datasets. In First Conference on Language Modeling.
>
>
>
> >**Q2. Safety trade-offs. If one suppresses the compliance direction to harden models against jailbreaks, what is the impact on desirable “policy-compliant” adherence (e.g., following benign safety instructions)? Is there measurable collateral damage to safe refusal behavior?**
>
>
> **A9.**
> First, we would like to clarify that our experiments do not directly implement the intervention you describe, namely suppressing the prompted-unique component (i.e. the compliance direction) and then re-measuring benign, policy-compliant safety behavior. In our work, we only amplify this component in order to understand how prompted value expression differs from intrinsic expression..
> That said, our results still provide indirect evidence about the likely trade-offs. When we steer with the prompted-unique component,    Taken together, these observations suggest that suppressing this direction would likely weaken prompt-level control in a fairly global way, reducing not only undesired compliance with adversarial personas but also desirable policy-compliant adherence under benign safety prompts.
>
> -----
> *For **Q3-6**, we believe that the core points have already been addressed in our earlier answers, and we indicate the corresponding references below.*
>
>
> >**Q3. Labeling robustness. Can you report cross-language, cross-value human agreement statistics with confidence intervals? How sensitive are results to the choice of evaluator model(s)? Do you observe systematic biases (e.g., over-penalizing or over-rewarding “safety” or “conformity”)?**
>
> **A10.**
> We believe this concern is largely addressed in **A1**.
>
>
> >**Q4. Statistical alignment of shared axes. Can you quantify the match to the Schwartz circle using formal statistics (e.g., circular correlation, Procrustes alignment) across layers/models and provide p-values vs. randomized baselines?**
>
> **A11.**
> This point is covered in **A4**, where we quantify the alignment between the shared value axes and Schwartz’s circumplex using formal geometric analyses (including comparisons against randomized baselines) across layers and models.
>
>
> >**Q5. Scope of the compliance channel. Does steering along the mean “prompted − intrinsic” direction improve adherence on standard instruction tasks (summarization, translation, extraction) without degrading faithfulness? Are there conflicts with refusal strategies on safety-related prompts?**
>
> **A12.**
> We address this question in **A3**, where we study steering along the mean “prompted − intrinsic” direction on standard instruction-following tasks.
>
>
> >**Q6. Decoding controls. Under length-matching and with temperature/sampling sweeps, does the intrinsic > prompted diversity advantage persist? A figure/table summarizing these robustness checks would be helpful.**
>
> **A13.**
> This issue is discussed in **A5**, where we run decoding-control experiments with metrics that consider length bias, and summarize whether the intrinsic > prompted diversity advantage persists under these settings.

---

### Official Review · Reviewer_1xKU · 2025-11-01

**Soundness:** 1
**Presentation:** 3
**Contribution:** 2
**Rating:** 4
**Confidence:** 4

**Summary:**

This paper mechanistically contrasts intrinsic and prompted value expression in LLMs.Using difference-in-means and vector orthogonalization, the authors isolate shared and unique components for both mechanisms.Intervention experiments show that while a shared component encodes general value semantics, the unique components diverge functionally.The intrinsic-unique component enhances lexical diversity, while the prompted-unique component is identified as a general "instruction following" mechanism.

**Strengths:**

1.Interesting question: The paper addresses an interesting question about value alignment mechanisms in LLMs.

2.Practical application: The findings may be useful for AI safety.

**Weaknesses:**

1.Limited theoretical framework: The paper lacks a clear theoretical explanation for why intrinsic and prompted mechanisms differ in these specific ways.

2.Relatively simple decomposition method: The SVD-based decomposition using only two vectors to identify shared and unique axes is relatively simple.

3.Insufficient jailbreaking evaluation: The jailbreaking experiments use only 500 samples per benchmark, limiting statistical power and preventing fair comparison with baselines. So, claiming to “match or exceed state-of-the-art” is premature.

4.Incomplete mechanistic understanding: The paper only studies what differs between mechanisms but provides limited insight into how and why they emerge in the LLMs.

5.Limited model coverage: Testing only two model families (Qwen and Llama) and three models. Lack of larger models, different model families and newer models.

6.Confounding from instruction-tuning: All tested models are instruction-tuned, which explicitly trains models to follow system prompts. The finding that "prompted-unique mechanisms strengthen instruction-following" may simply reflect instruction-tuning artifacts rather than fundamental properties of value expression. Without base model experiments, it's unclear whether these distinct mechanisms exist inherently or emerge specifically from instruction-tuning procedures.

**Questions:**

See weakness above.

---

> ### Author Response · Authors · 2025-11-25
> **Official Comment by Authors (Part 1/6)**
>
> Thank you for the constructive feedback and for recognizing the significance of our research question regarding value alignment mechanisms. We particularly appreciate the insightful comments on the potential confounding factors of instruction tuning and the need for broader model coverage, which have helped us significantly strengthen the soundness of our work.
>
> >**W1. Limited theoretical framework: The paper lacks a clear theoretical explanation for why intrinsic and prompted mechanisms differ in these specific ways.**
>
>
> **A1.** We agree that providing a theoretical grounding for the observed mechanistic differences is essential for a complete understanding. Below, we aim to connect our observations about the shared mechanism, the intrinsic-unique mechanism, and the prompted-unique mechanism with prior work, providing our interpretations of why these mechanisms may exhibit the reported behaviors.
>
> **Shared mechanism.** According to our experiments, the shared component captures general value-related semantics (Section 4.1).  We interpret this component as the core representation of value concepts established throughout the model's training pipeline. Theoretically, this is supported by mechanistic studies suggesting that high-level semantic features emerge during the pretraining phase [1][2] and the post-training processes (such as RLHF) [3]. Consequently, this shared mechanism serves as a necessary foundation: both intrinsic expression (reflecting the model’s internalized priors) and prompted expression (following explicit instructions) must access these same underlying latent concepts to produce value-consistent behaviors..
>
> **Intrinsic-unique mechanism.** According to our experiments, the intrinsic-unique component seems to be related to value expression using a more diverse vocabulary (Section 4.2). We see at least two complementary explanations for this pattern. First, during pretraining, the model is exposed to large-scale, naturalistic, instruction-free text, so it is likely to have learned value expressions across diverse discourse contexts and phrasings. As a result, the instruction-unique mechanism, without the influence of system prompts (even indirectly through the shared component), is likely to express values more freely and promote linguistic diversity. Second, the alignment phase during model training may also encourage diversity, as SFT models have been found to generate generic or overused responses, and alignment processes aim to promote more varied and informative outputs [4][5][6]. Overall, we believe these factors may promote language diversity in the intrinsic-unique mechanism.
>
> **Prompted-unique mechanism.** The prompted-unique component, in contrast, primarily enhances literal instruction-following and reuse of prompt-related keywords (Section 4.2). This pattern aligns closely with the objectives of RLHF-based alignment methods. Prior work on RLHF [7] has shown that alignment-stage supervision tends to encourage model behavior that closely follows explicit instructions and annotator-preferred formats. As a result, models may develop a strong tendency toward surface-level compliance, such as echoing instruction tokens or closely mirroring prompt phrasing. Mechanistically, this pattern is consistent with the emergence of ‘induction heads’ and related copying circuits, which attend to earlier occurrences of tokens and increase their logits, effectively implementing a copying algorithm [8][9].
> We thank the reviewer for providing us with the opportunity to elaborate on the connection between our observations and theoretical findings. We have incorporated this discussion in Appendix M.
>
> [1] Yang Xu, Yi Wang, and Hao Wang. Tracking the Feature Dynamics in LLM Training: A Mechanistic Study. arXiv preprint arXiv:2412.17626, 2024.
>
> [2] Siyu Chen, Heejune Sheen, Tianhao Wang, and Zhuoran Yang. Unveiling Induction Heads: Provable Training Dynamics and Feature Learning in Transformers. NeurIPS 2024.
>
> [3] Hongzhe Du, Weikai Li, Min Cai, Karim Saraipour, Zimin Zhang, Yizhou Sun, Himabindu Lakkaraju, and Shichang Zhang. How Post-Training Reshapes LLMs: A Mechanistic View on Knowledge, Truthfulness, Refusal, and Confidence. arXiv preprint arXiv:2504.02904, 2025.
>
> [4] Jiwei Li, Michel Galley, Chris Brockett, Jianfeng Gao, and Bill Dolan. A diversity-promoting objective function for neural conversation models. NAACL-HLT 2016, 2016.
>
> [5] Yizhe Zhang, Michel Galley, Jianfeng Gao, Zhiting Hu, Chris Brockett, and Bill Dolan. Generating informative and diverse conversational responses via adversarial information maximization. NeurIPS 2018, 2018.
>
> [6] Sanghwan Han, Byeongchang Kim, and Byoung-Tak Zhang. Measuring and improving semantic diversity of dialogue generation. Findings of EMNLP 2022, 2022.

---

> ### Author Response · Authors · 2025-11-25
> **(Part 2/6)**
>
> [7] Yuntao Bai, Andy Jones, Kamal Ndousse, Amanda Askell, Anna Chen, Nova DasSarma, Dawn Drain, Saurav Kadavath, Jackson Kernion, Tom Conerly, Sheer El-Showk, et al. Training a Helpful and Harmless Assistant with Reinforcement Learning from Human Feedback. arXiv preprint arXiv:2204.05862, 2022.
>
> [8] Charles Olsson, Nelson Elhage, Neel Nanda, Nicholas Joseph, Nova DasSarma, Tom Henighan, et al. In-context learning and induction heads. arXiv preprint arXiv:2209.11895, 2022.
>
> [9] Hongyi Wan, Hanyu Dong, Chengyu Wang, Xiaorui Liu, Zhihan Zhang, Daya Guo, Jiahai Wang, and Zhiyuan Liu. Tracing and Diagnosing Copying Circuits in Large Language Models. arXiv preprint arXiv:2402.02945, 2024.
>
> >**W2. Relatively simple decomposition method: The SVD-based decomposition using only two vectors to identify shared and unique axes is relatively simple.**
>
> **A2.** We agree that our SVD-based two-vector decomposition method is conceptually simple. Despite its simplicity, SVD effectively and efficiently serves our goal as follows.
>
> **(1) Geometric Intuition and Interpretability.**
> SVD provides a geometrically well-defined, intuitive, closed-form decomposition of the joint subspace aligned with our goal. Specifically, it isolates the one-dimensional direction that is maximally shared between intrinsic and prompted activations, and then separates this from the direction along which the two mechanisms differ.
>
> **(2) Alignment with Established Methodologies.**
> A substantial body of prior work has also used SVD to identify dominant shared directions or subspaces in neural representations, and our use of SVD follows this line of research. For example, SVCCA [10] first applies SVD to layer activations before comparing them across networks, and methods such as SEA [11] and SafeSteer [12] construct steering vectors from stacked activation vectors via SVD or PCA. In generative modeling, SeFa [13] uses an eigendecomposition of weight matrices to discover controllable semantic directions in latent space. Our two-vector SVD can be viewed as a minimal instance of such methods, specialized to the decomposition of intrinsic and prompted mechanisms that we study.
>
> **(3) Empirical Stability and Robustness.**
> Additionally, SVD yields stable decompositions that are robust to noise and sample variation. To verify this, we systematically evaluated the stability of the shared axis through a bootstrap analysis. Specifically, we repeatedly sampled 5% subsets from the extraction set (51 resamples), recomputed the shared and unique axes for each subset, and quantified stability in two ways: (1) by measuring the mean pairwise cosine similarity among bootstrapped shared directions, and (2) by comparing each bootstrapped direction to the original shared direction. The results are shown in the table below.
>
> | Model        | # Bootstraps | Mean pairwise cosine (95% CI) | Mean cosine to original (95% CI) |
> |-------------|--------------|-------------------------------|-----------------------------------|
> | Qwen2.5 7B  | 51           | 0.955 ± 0.004                 | 0.918 ± 0.011                     |
> | Qwen2.5 1.5B| 51           | 0.976 ± 0.002                 | 0.934 ± 0.007                     |
> | Llama3.1 8B | 51           | 0.955 ± 0.001                 | 0.971 ± 0.001                     |
>
> Empirically, we found that the shared axis is highly stable: the mean pairwise cosine similarity across bootstrap samples was consistently in the 0.95-0.98 range (with narrow 95% confidence intervals), and the average cosine similarity to the original direction was also high (≈0.92-0.97, depending on the model). This suggests that the identified shared direction is not an artifact of a particular sample, but rather a robust feature of the representation space. While our SVD-based decomposition is intentionally simple, it reliably recovers a stable shared/unique axis structure in practice, supporting the validity of our approach.
> We have incorporated the discussion on methodological justification in Appendix A.4.
>
> [10] M. Raghu, J. Gilmer, J. Yosinski, and J. Sohl-Dickstein. *SVCCA: Singular Vector Canonical Correlation Analysis for Deep Learning Dynamics and Interpretability*. NeurIPS, 2017.
>
> [11] Qiu et al., Spectral Editing of Activations for Large Language Model Alignment, NeurIPS 2024.
>
> [12] SafeSteer: Adaptive Subspace Steering for Efficient Jailbreak Defense in Vision-Language Models, arXiv:2509.21400, 2025.
>
> [13] Y. Shen and B. Zhou. *Closed-Form Factorization for Deep Generative Models (SeFa)*. CVPR, 2021.

---

> ### Author Response · Authors · 2025-11-25
> **(Part 3/6)**
>
> >**W3. Insufficient jailbreaking evaluation: The jailbreaking experiments use only 500 samples per benchmark, limiting statistical power and preventing fair comparison with baselines. So, claiming to “match or exceed state-of-the-art” is premature.**
>
> **A3.** Thank you for raising this point. We would like to clarify that the confusion was due to an error in our description. In appendix J, we stated that we sampled 500 queries per benchmark, but in fact, we used all available items: 520 queries for AdvBench and 400 text-only queries for HarmBench (excluding the multimodal subset). We apologize for any confusion this may have caused.
> That said, we agree that evaluating on approximately 500 queries can limit statistical power, especially when making comparisons to strong baselines. To address this, we ran each jailbreak evaluation five times with different random seeds (temperature 0.7) and reported 95% confidence intervals for attack success rates. The results are shown in the table below.
>
> | Target Model  & Benchmark      | GCG  | PAIR | TAP | DR  | Human | DSN  | OURS 95% CI    |
> |----------------------|------|------|-----|-----|-------|------|----------------|
> | Llama-3.1-8B-Instruct & Advbench | 58%  | 6%   | 2%  | 2%  | 1%    | 81%  | 97.2% ± 2.7%   |
> | Llama-3.1-8B-Instruct & Harmbench | --  | --  | --  | --  | --  | --  |90.4% ± 1.9%   |
> | Qwen2.5-7B-Instruct & Advbench  | 90%  | 34%  | 34% | 5%  | 70%   | 99%  | 89% ± 3%       |
> | Qwen2.5-7B-Instruct & Harmbench  | --  | --  | --  | --  | --  | --  |83% ± 2.1%   |
>
>
> We evaluated our method on both benchmarks, but for our comparative analysis we focus on AdvBench, as baseline results from the cited literature are primarily available for AdvBench. The results indicate that, within the confidence intervals, the prompted-unique component achieves higher success rates than other baselines with Llama-3.1-8B-Instruct. For Qwen2.5-7B-Instruct, the prompted-unique component improves over most baselines except DSN. In light of this, we agree that our original claim `matching or exceeding state-of-the-art jailbreak methods' is too strong. We have removed the original wording from Section 4.3
>
> >*...matching or exceeding state-of-the-art jailbreak methods…*
>
> In the revised manuscript, we instead describe the results more neutrally, for example:
>
> >*...consistent instruction-following capabilities across models, particularly achieving performance comparable to strong baselines on Llama-3.1-8B-Instruct.…*
>
> Please note that the prompted-unique direction component is not intended to outperform state-of-the-art jailbreak models. Rather, we are using jailbreaking tasks as additional evidence that the prompted-unique component functions as a general instruction-following mechanism, even in a distant task. We have included the updated analysis in the revised version of the manuscript (Table 21 in Appendix J).
>
>
> >**W4: Incomplete mechanistic understanding: The paper only studies what differs between mechanisms but provides limited insight into how and why they emerge in the LLMs.**
>
> **A4.**
> We believe that the discussion regarding the emergence and theoretical underpinnings of these mechanisms is largely covered in our response to **W1**. If you find that our discussion in **A1** is insufficient or if you have further questions regarding specific aspects of the mechanistic emergence, please let us know. We would be happy to provide additional clarifications to fully address your concerns.

---

> ### Author Response · Authors · 2025-11-25
> **(Part 4/6)**
>
> >**W5: Limited model coverage: Testing only two model families (Qwen and Llama) and three models. Lack of larger models, different model families and newer models.**
>
> **A5.** We agree that broader model coverage is important for assessing the robustness of our conclusions, and we have therefore conducted additional experiments to test generalization beyond the original three models. In particular, we now include larger and more diverse model families: Qwen2.5-32B-Instruct, Gemma2-9B-it, Qwen3-14B, and Qwen3-8B. Our choice of models is constrained by the fact that the *transformer_lens* toolkit that we used in our research currently supports only a subset of recent architectures. However, within this constraint, we selected models that vary in size, family, and training pipeline. The new experiments are reported in the revised manuscript under Appendix L, which we summarize below.
>
> ## Behavioral comparisons (steerability and diversity)
>
> **Steering effects.**
> We first evaluate value steering on the multilingual PVQ benchmark, using the same 6-point value rating scale as in the main paper and reporting mean changes relative to the unsteered model (see Section 3.2 for details).
>
> | Model | Mode | English | Chinese | Spanish | French | Korean | Avg |
> | :--- | :--- | :--- | :--- | :--- | :--- | :--- | :--- |
> | Qwen2.5-32B-Instruct | intrinsic | +0.91 | +0.46 | +0.47 | +0.34 | +1.14 | **+0.66** |
> | Qwen2.5-32B-Instruct | prompted | +1.17 | +0.95 | +0.88 | +0.86 | +1.47 | **+1.07** |
> | Gemma2-9b-it | intrinsic | +0.85 | +0.69 | +1.15 | +1.01 | +0.81 | **+0.90** |
> | Gemma2-9b-it | prompted | +2.11 | +1.03 | +2.24 | +2.03 | +1.66 | **+1.81** |
> | Qwen3-14B | intrinsic | +1.24 | +1.42 | +0.90 | +0.90 | +1.15 | **+1.12** |
> | Qwen3-14B | prompted | +1.67 | +1.36 | +1.45 | +0.47 | +0.94 | **+1.20** |
> | Qwen3-8B | intrinsic | +0.95 | +0.77 | +0.88 | +1.15 | +0.11 | **+0.77** |
> | Qwen3-8B | prompted | +1.27 | +0.66 | +1.96 | +1.59 | -0.96 | **+0.87** |
>
>
> Across all four models and five languages, we show that both intrinsic and prompted directions consistently increase PVQ scores, validating the general steerability of our value vectors. Consistent with our main results, prompted vectors generally yield stronger steering effects.
>
> **Response diversity.**
> We further examined the impact of steering on response diversity using lexical (Distinct-2/3, Entropy) and semantic (Embedding variation) metrics (see Section 3.3 for details). The following table shows the results for Qwen3-8B, averaged across all value dimensions. The comprehensive results of the remaining models are available in Appendix L.1.2.
>
>
> | Model                | Vector type         | Distinct-2 / 3 ↑ | Entropy-2 / 3 ↑  | Embedding variation ↑ | Frequently occurring words (Achievement) |
> |---------------------|------------------|------------------|------------------------|------------------------------------------|---------------------|
> | Qwen3-8B        | intrinsic           | **0.271 / 0.463**    | **11.750 / 13.090**  | **0.569** | personal, short, benefits |
> | Qwen3-8B        | prompted            | 0.169 / 0.286  | 10.047 / 11.36    | 0.479   | opportunities, risks, strategic |
> | Qwen3-14B        | intrinsic           | **0.396 / 0.671**    | **12.509 / 14.012**  | **0.498** | approach, potential, maintain |
> | Qwen3-14B         | prompted            | 0.296 / 0.537  | 11.784 / 13.474    | 0.438  |growth, personal, success |
> | Gemma2-9b-it         | intrinsic           | **0.430 / 0.718**    | **13.262 / 14.709**  | **0.569** | potential, market, financial  |
> | Gemma2-9b-it         | prompted            | 0.357 / 0.618  | 12.441 / 13.951    | 0.479   | position, success, embrace |
> | Qwen2.5-32B           | intrinsic             | **0.404 / 0.702**  | **12.970** / 14.312    | **0.526**   | industry, continuous, learning |
> | Qwen2.5-32B         | prompted          | 0.374 / 0.667    | 12.859 / **14.478** | 0.493 |  potential, benefits, plan |
>
> Intrinsic vectors consistently yield higher lexical diversity. Other metrics such as distinct-n and embedding variation also remain superior, with a minor deviation of entropy-3 in Qwen2.5-32B. In contrast, prompted vectors result in constrained diversity, frequently repeating explicit value-related terms (e.g., “opportunities”, "success", "growth"). This reinforces our core claim: intrinsic mechanisms facilitate varied, natural value expression, whereas prompted mechanisms operate by narrowing the output distribution toward specific, prompt-compliant keywords.

---

> ### Author Response · Authors · 2025-11-25
> **(Part 5/6)**
>
> ## Analysis of component roles
>
> **Shared components.**
> We extended the Schwartz-circle analysis to the new models by projecting shared axes of value vectors into the 2D plane of their principal components (see Section 4.1 for details). In all four cases, the intrinsic and prompted vectors converge onto a common set of shared directions that preserve the theoretical structure of Schwartz values (e.g., identifying "Universalism" and "Benevolence" as neighbors, distinct from opposing values like "Power"). This confirms that the shared component represents a context-general value representation that generalizes across model families.
>
> **Intrinsic-unique components.**
> We analyzed the vocabulary projections of the value vectors at the final layer (see Section 4.2. for details). Consistent with the main paper, intrinsic vectors in the new models promote semantically rich, context-dependent tokens (e.g., diversity, tolerance for Universalism), whereas prompted vectors prioritize formulaic, high-probability terms (e.g., respect, guidelines).
>
> We quantified this observation by computing the entropy of the post-softmax logits induced by the value vectors (see table below). Intrinsic-unique components consistently exhibit significantly higher entropy than prompted-unique components, supporting the conclusion that intrinsic mechanisms encode values through broader conceptual associations rather than narrow keyword matching.
>
>
> | Model                | prompted | intrinsic | prompted_orthogonal | intrinsic_orthogonal |
> |----------------------|----------|-----------|---------------------|----------------------|
> | Qwen2.5-32B-Instruct | 0.28     | 0.36      | 0.20                | **0.54**             |
> | Gemma2-9B-it         | 0.06     | 0.07      | 0.06                | **0.17**             |
> | Qwen3-14B            | 0.13     | 0.28      | 0.09                | **0.34**             |
> | Qwen3-8B             | 0.09     | 0.18      | 0.14                | **0.30**             |
> |......................|..........|...........|.....................|......................|
> | Mean                 | 0.14     | 0.22      | 0.12                | **0.34**             |
>
>
>
> **Prompted-unique components.**
> Finally, we tested the functional role of the prompted-unique component via jailbreaking experiments on AdvBench and HarmBench. As shown below, increasing the steering weight along the prompted-unique direction monotonically increases the Attack Success Rate (ASR) across all new models.
> | Model | Benchmark | Metric | Weight = 2 | Weight = 4 | Weight = 6 | Weight = 8 | Weight = 10 |
> | :--- | :--- | :--- | :--- | :--- | :--- | :--- | :--- |
> | **Qwen2.5-32B-Instruct** | AdvBench | ASR@1 | 0.12 | 0.21 | 0.37 | 0.58 | 0.76 |
> | | | ASR@9 | 0.45 | 0.63 | 0.80 | 0.92 | 0.96 |
> | | HarmBench | ASR@1 | 0.20 | 0.28 | 0.46 | 0.60 | 0.69 |
> | | | ASR@9 | 0.70 | 0.78 | 0.85 | 0.89 | 0.91 |
> | **Qwen3-8B** | AdvBench | ASR@1 | 0.01 | 0.19 | 0.24 | 0.29 | 0.32 |
> | | | ASR@9 | 0.03 | 0.51 | 0.59 | 0.65 | 0.80 |
> | | HarmBench | ASR@1 | 0.06 | 0.12 | 0.16 | 0.20 | 0.24 |
> | | | ASR@9 | 0.22 | 0.40 | 0.48 | 0.56 | 0.66 |
> | **Qwen3-14B** | AdvBench | ASR@1 | 0.12 | 0.13 | 0.44 | 0.77 | 0.90 |
> | | | ASR@9 | 0.47 | 0.38 | 0.82 | 0.95 | 0.99 |
> | | HarmBench | ASR@1 | 0.18 | 0.22 | 0.40 | 0.55 | 0.68 |
> | | | ASR@9 | 0.62 | 0.66 | 0.79 | 0.86 | 0.90 |
> | **Gemma2-9B-it** | AdvBench | ASR@1 | 0.18 | 0.22 | 0.28 | 0.36 | 0.49 |
> | | | ASR@9 | 0.51 | 0.58 | 0.67 | 0.77 | 0.86 |
> | | HarmBench | ASR@1 | 0.16 | 0.20 | 0.26 | 0.34 | 0.42 |
> | | | ASR@9 | 0.57 | 0.63 | 0.69 | 0.76 | 0.82 |
>
> For all models in both benchmarks, increasing steering strength of the prompted-unique component consistently increases jailbreak success rates. This shows that even in larger and architecturally different models, prompted-unique components reliably modulate instruction-following in adversarial settings.
>
> In summary, the new experiments show that (i) intrinsic and prompted steering improve value expression across larger and more diverse model families, (ii) intrinsic(-orthogonal) directions consistently yield higher response diversity and lexical entropy than prompted(-orthogonal) directions, and (iii) the shared, intrinsic-unique, and prompted-unique components exhibit the same qualitative behaviors in Schwartz-circle, lexical, and jailbreaking analyses as in the models we originally reported. We believe these results substantially strengthen the case that our decomposition captures general mechanisms rather than idiosyncrasies of a narrow model set. We have included the full details of these additional experiments in Appendix L of the revised manuscript. .

---

> ### Author Response · Authors · 2025-11-25
> **(Part 6/6)**
>
> >**W6. Confounding from instruction-tuning: All tested models are instruction-tuned, which explicitly trains models to follow system prompts. The finding that "prompted-unique mechanisms strengthen instruction-following" may simply reflect instruction-tuning artifacts rather than fundamental properties of value expression. Without base model experiments, it's unclear whether these distinct mechanisms exist inherently or emerge specifically from instruction-tuning procedures.**
>
>
> **A6.** Thank you for highlighting the importance of including base model experiments to disentangle potential artifacts from instruction-tuning In response to your feedback, we conducted supplementary analyses using a base model without instruction-tuning (Qwen2.5-7B). After extracting both intrinsic and prompted value vectors, we examined whether steering with the prompted-unique component still influences instruction-following behavior in a jailbreak setting (Advbench).
>
>
> | metric \ steering coefficient | -10    | -4     | 0 (base) | 4      | 10     |
> |-----------------|--------|--------|----------|--------|--------|
> | Attack Success Rate(ASR) (%)         | 56.52  | 74.29  | 89.47    | 96.84  | 97.27  |
> | ΔASR (pp)       | -32.95 | -15.17 | 0.00     | +7.37  | +7.80  |
>
> We find that while the base model already exhibits a relatively high rate of jailbreak responses (89.47%), which is expected because it did not go through an alignment phrase, steering with the prompted-only component further increases jailbreak success rates substantially (97.27%), and applying negative steering substantially decreases the rate of jailbreak completions (56.52%). These results indicate that, even without explicit instruction-tuning, the prompted-unique component we identify still affects the model’s compliance to system prompts. This suggests that the prompted-unique pathway is not purely an artifact of instruction-tuning, but instead reflects a more general instruction-following mechanism already present in the base model. We have incorporated these experimental results in the revised version of our manuscript. Thank you again for your constructive suggestion.

---

> > ### Comment · Reviewer_1xKU · 2025-11-27
> > **Response to Author**
> >
> > Thank you for your effort and detailed rebuttal. It's ok with questions 2, 3, 5, and 6.
> >
> > However, I still have some reservations about the theoretical framework and the definitions of "Intrinsic" and "Prompted" values.
> >
> > 1.  **Ambiguous Definition of "Value":** In the context of AI alignment, a "value" is typically understood as a scalar preference or reward. The paper treats it as a vector direction without sufficient theoretical justification for why this representation is valid.
> > 2.  **Conflation with Instruction Following:** The "Prompted" mechanism appears to be mechanistically equivalent to generic instruction following or lexical priming. As noted in your rebuttal, this mechanism relies on "induction heads" and "copying algorithms" to increase the logits of prompt-related tokens. Labeling a mechanism that primarily copies keywords as a distinct "Value Mechanism" feels conceptually stretched.

---

> ### Author Response · Authors · 2025-11-28
> **Response to reviewer 1xKU (Part 1/3)**
>
> Ah, thanks for the clarification. This helps us determine where to begin, and below we provide additional theoretical and empirical background to situate our work.
>
> >**1. Ambiguous Definition of "Value": In the context of AI alignment, a "value" is typically understood as a scalar preference or reward. The paper treats it as a vector direction without sufficient theoretical justification for why this representation is valid.**
>
> **A1.**
> As you mentioned, in many AI alignment settings, researchers have selected specific values (typically helpfulness, harmlessness, and honesty) and represented this “collective value” as a single scalar preference or reward. However, researchers have pointed out that real-world human values are far more diverse, inherently pluralistic, and sometimes conflicting (e.g., safety vs. freedom). In response, they have recently sought to incorporate more diverse and pluralistic values into AI systems [1, 2]. In this context, Schwartz’s theory of basic values, which defines ten universal human values, has been widely adopted in AI research [3, 4, 5] as a concrete and extensively validated value taxonomy.
>
> One challenge is how to align LLMs with multiple values simultaneously, since most alignment algorithms rely on a single scalar value score. As a result, researchers have turned to value steering: an inference-time method for steering a trained model toward specific values. This method is more cost-efficient than full training and more flexible, as it can control multiple values simultaneously. A widely used steering paradigm is to manipulate LLM hidden states under the assumption that different values reside as different directions in activation space. A theoretical foundation for this assumption is the linear representation hypothesis, which posits that various features and concepts in the world are encoded in approximately linear subspaces (e.g., the well-known example from word2vec: king – man + woman = queen) [6].
>
> *But does this extend to abstract features like human values?* A growing body of work suggests that features such as human values, personality traits, political orientation, emotions, and style can be monitored and steered via linear directions in activation space [7, 8, 9, 10]. Our work also supports this view by demonstrating that vector steering can intensify or reduce the expression of certain values, and that the hidden state space in certain layers reflects the theoretical circular structure of values proposed by Schwartz (i.e., similar values appear close together, and opposing values lie on opposite sides) (Figure 7). Our representation of values as vectors in LLMs’ activation spaces is informed by this literature.
>
> *Why is this theoretically plausible?* Although there is currently no theoretical guarantee yet, the existence of such vectors is plausible because different values are associated with somewhat distinct vocabularies. We can therefore think of specific directions in hidden states as promoting the logits of value-associated vocabularies, directions that the model could have learned during training. To develop a more robust theoretical foundation, further research is needed to analyze the behavior and underlying mechanisms of these “value directions”, which our work aims to advance.
>
> ----
>
> ###### [1] Sorensen, T., Moore, J., Fisher, J., Gordon, M. L., Mireshghallah, N., Rytting, C. M., ... & Choi, Y. (2024, July). Position: A Roadmap to Pluralistic Alignment. In International Conference on Machine Learning (pp. 46280-46302). PMLR.
>
> ###### [2] Rozen, N., Bezalel, L., Elidan, G., Globerson, A., & Daniel, E. Do LLMs have Consistent Values?. In The Thirteenth International Conference on Learning Representations.
>
> ###### [3] Yao, J., Yi, X., Gong, Y., Wang, X., & Xie, X. (2024, June). Value FULCRA: Mapping large language models to the multidimensional spectrum of basic human value. In Proceedings of the 2024 Conference of the North American Chapter of the Association for Computational Linguistics: Human Language Technologies (Volume 1: Long Papers) (pp. 8762-8785).
>
> ###### [4] Shen, H., Knearem, T., Ghosh, R., Yang, Y. J., Clark, N., Mitra, T., & Huang, Y. (2025, November). ValueCompass: A Framework for Measuring Contextual Value Alignment Between Human and LLMs. In Proceedings of the 9th Widening NLP Workshop (pp. 75-86).
>
> ###### [5] Ye, H., Xie, Y., Ren, Y., Fang, H., Zhang, X., & Song, G. (2025, April). Measuring human and ai values based on generative psychometrics with large language models. In Proceedings of the AAAI Conference on Artificial Intelligence (Vol. 39, No. 25, pp. 26400-26408).
>
> ###### [6] Park, K., Choe, Y. J., & Veitch, V. (2024, July). The Linear Representation Hypothesis and the Geometry of Large Language Models. In International Conference on Machine Learning (pp. 39643-39666). PMLR.

---

> ### Author Response · Authors · 2025-11-28
> **(Part 2/3)**
>
> ###### [7] Kim, J., Evans, J., & Schein, A. Linear Representations of Political Perspective Emerge in Large Language Models. In The Thirteenth International Conference on Learning Representations.
>
> ###### [8] Chen, R., Arditi, A., Sleight, H., Evans, O., & Lindsey, J. (2025). Persona vectors: Monitoring and controlling character traits in language models. arXiv preprint arXiv:2507.21509.
>
> ###### [9] Konen, K., Jentzsch, S., Diallo, D., Schütt, P., Bensch, O., El Baff, R., ... & Hecking, T. (2024, March). Style Vectors for Steering Generative Large Language Models. In Findings of the Association for Computational Linguistics: EACL 2024 (pp. 782-802).
>
> ###### [10] Jin, H., Li, M., Wang, X., Xu, Z., Huang, M., Jia, Y., & Lian, D. (2025, July). Internal Value Alignment in Large Language Models through Controlled Value Vector Activation. In Proceedings of the 63rd Annual Meeting of the Association for Computational Linguistics (Volume 1: Long Papers) (pp. 27347-27371).
>
> ---
>
> >**2. Conflation with Instruction Following: The "Prompted" mechanism appears to be mechanistically equivalent to generic instruction following or lexical priming. As noted in your rebuttal, this mechanism relies on "induction heads" and "copying algorithms" to increase the logits of prompt-related tokens. Labeling a mechanism that primarily copies keywords as a distinct "Value Mechanism" feels conceptually stretched.**
>
>
>
> **A2.**
> We believe the confusion potentially stems from an unclear distinction between what we call the *prompted value mechanism* and the *prompted-only component* (since the “induction heads” and “copying algorithms” you mentioned pertain to the latter, not the former), and we will clarify this here. We understand that one can view the prompted value mechanism (i.e., “following a system prompt to elicit a specific value” in its entirety) as a generic instruction-following mechanism. However, an alternative, more granular interpretation (or hypothesis) is that this behavior can be decomposed into two distinct components: (1) expression of the specified value, and (2) pure instruction compliance. We clarify below what each entails and how this hypothesis is supported by prior research (though it is not yet definitively established).
>
> (1) **Value expression component**: An LLM can naturally express a certain value without an explicit instruction. Accordingly, certain directions in the hidden states are highly likely to control the expression of specific values. Both our work and prior studies have shown that LLMs change the values they express when steering vectors are added to hidden states, even without prompt instructions [1, 2].
>
> (2) **Instruction compliance component**: This (somewhat abstract) component is hypothesized to control the model’s general “obedience” to instructions, beyond specific value expression. It could increase compliance in several ways: by suppressing alignment mechanisms (e.g., safety alignment), copying instruction tokens or related tokens, reducing questioning or hedging, discouraging the proposal of alternative views, and so on. *Is this a reasonable hypothesis?* Below is a summary of prior studies that offer supporting evidence:
>
>
> - *Anti-alignment mechanisms*: Vectors that control refusal behavior [3, 4], and features in hidden states that promote abstention for uncertain questions (e.g., “I don’t know”) [5].
>
> - *Induction and copying heads* [6, 7]: Specific directions in the attention or residual space that implement token copying and induction.
>
> - *Reducing questions or hedging*: Vectors that regulate levels of politeness, directness [8, 9], and confidence [10, 11].
>
> - *Discouraging alternative views*: Vectors that promote sycophantic behavior [12, 13].
>
> Experiment results based on our decomposition of the prompted value component into the *intrinsic-prompted shared component* and the *prompted-only component* reflect this hypothesis. Specifically, the shared component seems to correspond to value expression, as it exhibits a stronger value steering effect than both the intrinsic-only and prompted-only components (Figure 25). The prompted-only component seems to reflect instruction compliance (not all of the above behaviors, but to some extent), as it is effective for jailbreaking tasks that are not tied to specific values. Additionally, steering with the prompted value vector produces responses that are less diverse and shorter (9.7% shorter on average) than steering with the intrinsic value vector. This might reflect the influence of the instruction compliance component in the prompted value vector, which tends to promote copying prompt-related tokens and suppress the degree of freedom in generated text.

---

> ### Author Response · Authors · 2025-11-28
> **(Part 3/3)**
>
> Again, while this hypothesis remains unproven (yet), growing empirical evidence supports it. We believe it is important to further investigate these phenomena. **Our work contributes to this effort by examining the commonalities and differences between intrinsic and prompted value mechanisms, which could ultimately lead to a more comprehensive theoretical framework.**
>
> ###### [1] Chen, R., Arditi, A., Sleight, H., Evans, O., & Lindsey, J. (2025). Persona vectors: Monitoring and controlling character traits in language models. arXiv preprint arXiv:2507.21509.
>
> ###### [2] Deng, J., Tang, T., Yin, Y., Zhao, X., & Wen, J. R. Neuron based Personality Trait Induction in Large Language Models. In The Thirteenth International Conference on Learning Representations.
>
> ###### [3] Rimsky, N., Gabrieli, N., Schulz, J., Tong, M., Hubinger, E., & Turner, A. (2024, August). Steering llama 2 via contrastive activation addition. In Proceedings of the 62nd Annual Meeting of the Association for Computational Linguistics (Volume 1: Long Papers) (pp. 15504-15522).
>
> ###### [4] Arditi, Andy, et al. "Refusal in language models is mediated by a single direction." Advances in Neural Information Processing Systems 37 (2024): 136037-136083.
>
> ###### [5] Lindsey, et al., "On the Biology of a Large Language Model", Transformer Circuits, 2025.
>
> ###### [6]Olsson, C., Elhage, N., Nanda, N., Joseph, N., DasSarma, N., Henighan, T., ... & Olah, C. (2022). In-context learning and induction heads. arXiv preprint arXiv:2209.11895.
>
> ###### [7] Merullo, J., Eickhoff, C., & Pavlick, E. (2024). Talking heads: Understanding inter-layer communication in transformer language models. Advances in Neural Information Processing Systems, 37, 61372-61418.
>
> ###### [8] Konen, K., Jentzsch, S., Diallo, D., Schütt, P., Bensch, O., Baff, R., Opitz, D., Hecking, T. (2024, March). Style Vectors for Steering Generative Large Language Models. In Findings of the Association for Computational Linguistics: EACL 2024 (pp. 782-802).
>
> ###### [9] van der Weij, T., Poesio, M., & Schoots, N. (2024). Extending activation steering to broad skills and multiple behaviours. arXiv preprint arXiv:2403.05767.
>
> ###### [10] Venhoff, C., Arcuschin, I., Torr, P., Conmy, A., & Nanda, N. (2025). Understanding reasoning in thinking language models via steering vectors. arXiv preprint arXiv:2506.18167.
>
> ###### [11] Rahn, N., D'Oro, P., & Bellemare, M. G. (2024). Controlling large language model agents with entropic activation steering. arXiv preprint arXiv:2406.00244.
>
> ###### [12] Wang, M., la Tour, T. D., Watkins, O., Makelov, A., Chi, R. A., Miserendino, S., ... & Mossing, D. (2025). Persona features control emergent misalignment. arXiv preprint arXiv:2506.19823.
>
> ###### [13] Min, P. P., Paudel, A., Adityo, N., Zhu, A., Rufail, A., Blondin, C., ... & O'Brien, S. Mitigating Sycophancy in Language Models via Sparse Activation Fusion and Multi-Layer Activation Steering. In Mechanistic Interpretability Workshop at NeurIPS 2025.
>
> ---
>
> We are grateful for your follow-up and pleased to hear that our additional experiments and clarifications have resolved your concerns regarding W2, W3, W5, W6. If you have any further questions or requests, please let us know. We will make our best effort to address them.

---

> > ### Comment · Reviewer_1xKU · 2025-11-28
> >
> > We thank the author for the detailed response and the efforts they made. I'm sorry that the edit is gone. I'll raise my score to 6 once it's back. My concerns are addressed.

---

> > > ### Author Response · Authors · 2025-12-03
> > > **Thank you for raising the score.**
> > >
> > > We sincerely appreciate your constructive feedback and your decision to raise the score to 6.
> > >
> > > We are glad that our responses addressed your concerns. Thank you again for your time and support to improve our paper.

---

### Author Response · Authors · 2025-12-03
**General Response to All Reviewers (Part 1/3)**

We thank the reviewers for their careful review and constructive feedback. Since the scores have been reverted to their original ones, we would first like to highlight the score changes made by the reviewers before the system reverted them. We note that while Reviewers NTL2 and 1xKU raised their scores following our rebuttal, unfortunately, the system restrictions prevented us from discussing with Reviewers cWMP and DfyZ

| Reviewer | Original Score | Revised Score (Pre-Reversion) | Note |
| :--- | :--- | :--- | :--- |
| **Reviewer NTL2** | 6 | **8** | Concerns resolved |
| **Reviewer 1xKU** | 4 | **6** | Concerns resolved |
| **Reviewer cWMP** | 8 | 8 | System closed before rebuttal|
| **Reviewer DfyZ** | 4 | 4 | System closed before rebuttal |

---

Reviewers highlighted several strengths of the work.
### 1. Conceptual and mechanistic framing.
Reviewer NTL2 wrote that our work “presents an **interpretable approach** to analyzing the internal differences between no-prompt and prompted model behaviors” and that our paper “offers **concrete takeaways that deepen our understanding** of how adding prompts alters internal representations in LLMs”. The reviewer especially emphasized that the “integration of CAVs with MLP neuron analysis is **well executed and conceptually appealing**—using CAVs to interpret neuron activations is both **novel and insightful**.”

Reviewer cWMP noted that “This paper presents a **novel mechanistic analysis of LLM values**, in contrast to most related papers, which
conduct only behavioral evaluations” and that there are “**interesting empirical findings**, such as prompted > intrinsic in steerability; intrinsic > prompted in lexical diversity”.

### 2. Alignment and safety relevance.
Reviewer DfyZ noted  that “the work has **immediate value for both controllable generation and safety alignment**”. Similarly, reviewer cWMP remarked that “It holds **practical implications** for alignment and persona steering (e.g., steerability vs. diversity trade-offs; jailbreak effects)”, and also reviewer 1xKU noted that ”the paper addresses an **interesting question** about value alignment mechanisms in LLMs” and that “the findings may be **useful for AI safety**”.

### 3. Empirical scope and methodology.
Reviewer DfyZ emphasized that “The empirical study is **thorough**. It spans multiple model families and scales, multiple languages, and diverse behavioral evaluations”. The reviewer also highlighted that “The convergence of evidence -- from geometric similarities, behavioral steering outcomes, diversity metrics, and logit-lens vocabulary projections -- creates a **compelling narrative**.”

---

> ### Author Response · Authors · 2025-12-03
> **(Part 2/3)**
>
> # Revision Summary
>
> We have conducted **all the additional experiments** requested by the reviewers and incorporated them into the revised manuscript. Our revisions focus on five themes that directly address the main points raised across reviews as follows.
>
> ---
>
> ### 1. Are value directions real, stable, and theoretically grounded? (1xKU, DfyZ, NTL2)
> **Concern.** Reviewers asked whether our value directions genuinely reflect linear structure (rather than random artifacts), how they relate to Schwartz’s circumplex, and how they connect to prior work on linear representations and RLHF.
>
> **What we did.**  We added a bootstrap stability analysis confirming that the shared axis is highly stable across resamples (mean cosine > 0.95). We introduced formal geometric tests (e.g., Procrustes analysis) which verify that shared axes recover the Schwartz circumplex structure significantly better than randomized baselines. Additionally, our span-based and linear-probe comparisons demonstrate that our global difference-in-means vectors yield stronger and more consistent steering effects than span-restricted vectors or linear probes. Finally, we expanded Appendix M to connect our components to prior work on linear representations, RLHF, and copying circuits.
>
> ---
>
> ### 2. How broadly do the value-steering results generalize across models and languages? (DfyZ, cWMP)
>
> **Concern.** Reviewers asked whether our conclusions hold beyond the original three models and two families, and whether the intrinsic/prompted patterns are robust across architectures and languages.
>
> **What we did.** We added four more instruction-tuned models (Qwen2.5-32B-Instruct, Gemma2-9B-it, Qwen3-14B, Qwen3-8B) and evaluated intrinsic and prompted steering on the multilingual PVQ benchmark in five languages, together with extended diversity and component-role analyses. We found that for all additional models, both intrinsic and prompted vectors reliably increase PVQ scores, and the shared / intrinsic-unique / prompted-unique decomposition exhibits the same qualitative behavior as in the original models.
>
> ---
>
> ### 3. What is the true scope of the compliance pathway and our jailbreak claims? (DfyZ W2, W3; 1xKU)
>
> **Concern.** Reviewer DfyZ asked whether the mean ‘prompted − intrinsic’ direction is a general instruction-following booster on unrelated tasks, and raised concerns about the statistical strength of our jailbreak comparisons; Reviewer 1xKU also asked whether the compliance effect is an artifact of instruction-tuning.
>
> **What we did.**
>
> - For **jailbreaking**, we clarified that, in our original manuscript, we had already used all available AdvBench and HarmBench text items, and we had re-run each evaluation five times with different random seeds (temperature 0.7), reporting mean Attack Success Rates with 95% confidence intervals and per-prompt robustness metrics;
>
> - For **model and tuning dependence**, we evaluated steering with the prompted-unique direction on an untuned base model (Qwen2.5-7B) in the jailbreak setting, and showed that the same direction still modulates jailbreak success rates, indicating that the compliance component is not purely an artifact of instruction-tuning.
>
> - For **general instruction compliance** (DfyZ W2), we  evaluated the direction on two non-value benchmarks: GAMBIT (translation with gender constraints) and IFEVAL (atomic constraints). We found that steering enhances compliance when the target behavior is within the model's capabilities (GAMBIT), but acts as a modulator rather than a universal booster for low-level constraints (IFEVAL), helping to establish the functional boundaries of the compliance channel.

---

> ### Author Response · Authors · 2025-12-03
> **(Part 3/3)**
>
> ### 4. Do the mechanistic roles of shared vs. unique components hold beyond lexical priming? (NTL2 W1, DfyZ)
>
> **Concern.** Reviewer NTL2 requested concrete neuron-level analysis to substantiate our shared / intrinsic / prompted interpretation, and Reviewer DfyZ was concerned that diversity effects might be confounded by prompt length or explicit value keywords.
>
> **What we did.** We conducted additional experiments to perform automated neuron explanations for shared, intrinsic-unique, and prompted-unique neuron groups, and as a result identified many interpretable neurons whose behaviors align with our claims: shared neurons encode abstract value semantics, intrinsic-unique neurons respond to contextual cues that evoke the value, and prompted-unique neurons track explicit value definitions and canonical keywords.
>
> We also conducted additional analyses and showed that our diversity analysis is consistent across length-matched diversity metrics and temperature / top p sweeps. Lastly, we introduced implicit, keyword-free value  prompts that remove explicit keywords. We showed that prompted value vectors extracted with those implicit prompts still show the same trends as our original analysis (prompted vector steering less diverse than intrinsic vector steering).
>
> ---
>
> ### 5. Are our labels and evaluations reliable and statistically robust? (DfyZ)
> **Concern.** Reviewer DfyZ questioned whether our results might depend too heavily on a single judge model and noisy labels, and asked for stronger evidence of reliability.
>
> **What we did.** We extended human evaluation to 200 additional items with three annotators, reporting Fleiss’ $\kappa$, per-value agreement, and confidence intervals. We also re-ran the situational dilemmas experiment with multiple judge models (Qwen2.5-72B-Instruct, Qwen3-Next-80B-Instruct, gpt-4.1-mini). As a result, we observed moderate inter-annotator and inter-judge agreement and consistent trends for both intrinsic and prompted steering.
>
> We also updated Section 5 and the related-work discussion to cover recent SAE-based value and refusal-steering methods and other activation-steering approaches, in line with Reviewer 1xKU’s comments. We hope these revisions address the main concerns while preserving the strengths highlighted above, and we remain happy to revise further if the AC or reviewers see any remaining gaps.

---

### Note · Program_Chairs · 2026-01-17
**Submission Desk Rejected by Program Chairs**

The following references in this submission do not refer to real documents and/or have major errors in bibliographic information:

 Alex Turner, Andrew Scherlis, Kshitij Ghasem-Pour, Jacob Steinhardt, and Finale Doshi-Velez. Activation addition: A simple and scalable method for controlling large language models. arXiv preprint arXiv:2308.10248, 2023. URL https://arxiv.org/abs/2308.10248.

Hongyi Wan, Hanyu Dong, Chengyu Wang, Xiaorui Liu, Zhihan Zhang, Daya Guo, Jiahai Wang, and Zhiyuan Liu. Tracing and diagnosing copying circuits in large language models. arXiv preprint arXiv:2402.02945, 2024.